# Optimistic Mirror Descent Either Converges to Nash or to Strong Coarse Correlated Equilibria in Bimatrix Games

**Ioannis Anagnostides**
Carnegie Mellon University
Pittsburgh, PA 15213
ianagnos@cs.cmu.edu

**Gabriele Farina**
Carnegie Mellon University
Pittsburgh, PA 15213
gfarina@cs.cmu.edu

**Ioannis Panageas**
University of California Irvine
Irvine, CA 92697
ipanagea@ics.uci.edu

**Tuomas Sandholm**
Carnegie Mellon University
Strategy Robot, Inc.
Optimized Markets, Inc.
Strategic Machine, Inc.
Pittsburgh, PA 15213
sandholm@cs.cmu.edu

## Abstract

We show that, for any sufficiently small fixed $\epsilon > 0$, when both players in a *general-sum two-player (bimatrix) game* employ *optimistic mirror descent (OMD)* with smooth regularization, learning rate $\eta = O(\epsilon^2)$ and $T = \Omega(\text{poly}(1/\epsilon))$ repetitions, either the dynamics reach an $\epsilon$-*approximate Nash equilibrium (NE)*, or the average correlated distribution of play is an $\Omega(\text{poly}(\epsilon))$-*strong coarse correlated equilibrium (CCE)*: any possible unilateral deviation does not only leave the player worse, but will decrease its utility by $\Omega(\text{poly}(\epsilon))$. As an immediate consequence, when the iterates of OMD are bounded away from being Nash equilibria in a bimatrix game, we guarantee convergence to an *exact* CCE after only $O(1)$ iterations. Our results reveal that uncoupled no-regret learning algorithms can converge to CCE in general-sum games remarkably faster than to NE in, for example, zero-sum games. To establish this, we show that when OMD does not reach arbitrarily close to a NE, the *(cumulative) regret* of *both* players is not only *negative*, but *decays linearly* with time. Given that regret is the canonical measure of performance in online learning, our results suggest that cycling behavior of no-regret learning algorithms in games can be justified in terms of efficiency.

## 1   Introduction

The last few years have seen a tremendous amount of progress in computational game solving, witnessed over a series of breakthrough results on benchmark applications in AI [Bowling et al., 2015, Brown and Sandholm, 2017, Moravčík et al., 2017, Silver et al., 2016, Vinyals et al., 2019]. Most of these advances rely on algorithms for approximating a *Nash equilibrium (NE)* [Nash, 1950] in *two-player zero-sum games*. Indeed, in that regime it is by now well-understood how to compute a NE at scale. However, many real-world interactions are not zero-sum, but instead have *general-sum utilities*, and often more than two players. In such settings, Nash equilibria suffer from several drawbacks. First, finding even an approximate NE is *computationally intractable* [Daskalakis et al., 2008, Etessami and Yannakakis, 2007, Chen et al., 2009, Rubinstein, 2016]—subject to well-believed complexity-theoretic assumptions. Furthermore, even if we were to reach one, NE outcomes can be dramatically more inefficient in terms of the *social welfare* compared to other more permissive equilibrium concepts [Moulin and Vial, 1978]. Finally, NE suffer from *equilibrium selection* issues:

there can be a multitude of equilibria, and an equilibrium strategy may perform poorly against the "wrong" equilibrium strategies [Harsanyi et al., 1988, Harsanyi, 1995, Matsui and Matsuyama, 1995], thereby necessitating some form of communication between the players.

A competing notion of rationality is Aumann's concept of *correlated equilibrium (CE)* [Aumann, 1974], generalizing Nash's original concept. Unlike NE, a correlated equilibrium can be computed *exactly* in polynomial time [Papadimitriou and Roughgarden, 2008, Jiang and Leyton-Brown, 2015]. Further, CE can arise from simple *uncoupled learning dynamics*, overcoming the often unreasonable assumption that players have perfect knowledge over the game's utilities. As such, correlated equilibria constitute a much more plausible outcome under independent agents with *bounded rationality*. Indeed, it is folklore that when all players in a general game employ a *no-regret learning algorithm* [Hart and Mas-Colell, 2000], the average correlated distribution of play converges to a *coarse correlated equilibrium (CCE)*—a further relaxation of CE [Moulin and Vial, 1978].

Now as it happens, there are specific classes of games, such as *strictly competitive games* [Adler et al., 2009] or *constant-sum polymatrix games* [Daskalakis and Papadimitriou, 2009a, Cai and Daskalakis, 2011, Cai et al., 2016], for which CCE "collapse" to NE. In fact, although computing Nash equilibria in such games is amenable to linear programming [Adler, 2013], the state of the art algorithms are based on uncoupled learning procedures, for reasons mostly relating to scalability. Nevertheless, such algorithms require $\Omega(\text{poly}(1/\epsilon))$ iterations to reach an $\epsilon$-approximate Nash equilibrium [Daskalakis et al., 2011], meaning that the convergence is slow, particularly in the high precision regime—both in theory and in practice. Furthermore, prior literature treats convergence to NE in, for example, zero-sum games analogously to convergence to CCE in general-sum games. While this unifying treatment—which is an artifact of the no-regret framework—may seem compelling at first glance, it is unclear whether CCE share similar convergence properties to NE. Indeed, as we have alluded to, those equilibrium concepts are fundamentally different (in general games).

Our primary contribution is to challenge this traditional framework, establishing that *convergence to CCE can be remarkably faster than the convergence to NE*; this fundamental difference is showcased in Figure 1. In fact, we show that the only obstacle for converging *exactly* after only $O(1)$ iterations to a CCE in general-sum two-player games is reaching arbitrarily close to a NE. Our results also reveal an intriguing complementarity: the farther the dynamics are from NE, the faster the guarantee of convergence to CCE.

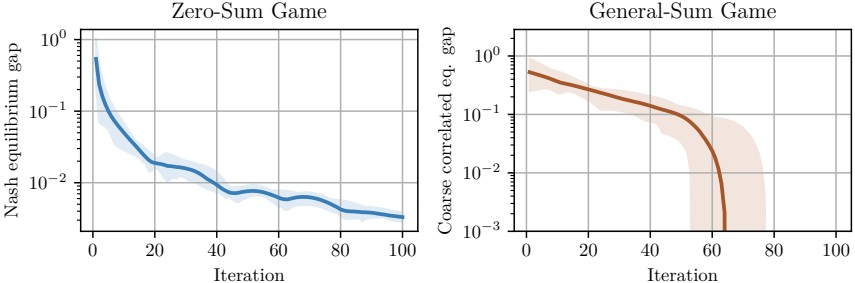

**Figure 1:** Convergence of uncoupled no-regret dynamics to NE in a zero-sum game (left image) versus convergence to CCE in a general-sum game (right image); further details are provided in Section 4.1.

## 1.1 Our Contributions

We study uncoupled no-regret learning dynamics in the fundamental class of two-player general-sum games (a.k.a. *bimatrix games*). Specifically, we focus on *optimistic mirror descent (OMD)* [Chiang et al., 2012, Rakhlin and Sridharan, 2013a], a variant of the standard mirror descent method which incorporates a *prediction* into the optimization step. Before we state our main result, let us introduce *strong CCE*, a *refinement* of CCE for which any unilateral deviation from a player is not only worse, but *decreases* its utility by an additive $\epsilon \geq 0$ (see Definition 2.4 for a formal description). In particular, we clarify that any strong CCE is (trivially) an *exact* CCE. We are now ready to state our main theorem.[1]

---

[1] For simplicity in the exposition of our results, we use the $O(\cdot)$ and $\Omega(\cdot)$ notation in the main body to suppress parameters that depend on the underlying game; precise statements are given in Appendix A.

**Theorem 1.1** (Abridged; Full Version in Corollary A.2). *Fix any sufficiently small $\epsilon > 0$, and suppose that both players in a bimatrix game employ OMD with learning rate $\eta = O(\epsilon^2)$ and smooth regularizer for $T = \Omega(\mathrm{poly}(1/\epsilon))$ repetitions. Then,*

- *Either the dynamics reach an $\epsilon$-approximate Nash equilibrium;*

- *Or, otherwise, the average correlated distribution of play is an $\Omega(\mathrm{poly}(\epsilon))$-strong CCE.*

Here, the convergence to Nash equilibrium is implied in a *last-iterate* sense (and not a typical time-average): *i.e.*, there exists a time $t \in [\![T]\!]$ for which the dynamics reach an $\epsilon$-approximate Nash equilibrium. Of course, in light of well-established impossibility results, OMD is certainly not going to reach an $\epsilon$-approximate NE—for a sufficiently small *constant* $\epsilon > 0$ [Rubinstein, 2016]—in *every bimatrix game*. As a result, an immediate interesting implication of Theorem 1.1 is that, by setting $\epsilon$ to be sufficiently small, OMD either yields the best known polynomial-time approximation for Nash equilibria in bimatrix games, or, otherwise, $O(1)$ iterations suffice to obtain a strong CCE. It is intriguing that the farther the dynamics are from yielding a Nash equilibrium, the *stronger* the CCE predicted by Theorem 1.1. Indeed, the only impediment for reaching a strong CCE after only $O(1)$ iterations lies in reaching very close to a Nash equilibrium.[2]

To establish Theorem 1.1, we prove that when the iterates of OMD are bounded away from being Nash equilibria, the regret of *both* players is not only negative, but *decreases linearly* over time (Theorem 3.4); see Figure 3 for an illustration. In this way, our approach represents a substantial departure from prior work which endeavored to characterize no-regret learning dynamics from a unifying standpoint for all possible games. Furthermore, besides the folklore connection of regret with CCE (which allows us to establish Theorem 1.1), regret is the canonical measure of performance in online learning. Hence, our results can be construed as follows: when the dynamics do not reach a Nash equilibrium, we obtain remarkably stronger performance guarantees. This seems to suggest that *cycling behavior in games may not be a bug, but a feature*.[3]

From a technical standpoint, a central ingredient in our proof is a remarkable property—discovered in prior works—coined as the `RVU` bound [Syrgkanis et al., 2015, Rakhlin and Sridharan, 2013a]. Specifically, our argument crucially leverages the rather enigmatic negative term in the `RVU` bound (Proposition 2.1), which was treated as a "cancellation factor" in prior works. This allows us to establish (conditionally) linear decay for the regrets in Theorem 3.4. To this end, an important component of our proof is to connect the path lengths of the two players in Lemma 3.3 through a potential-type argument; notably, such properties break in arbitrary games with more than two players (see Remark 3.7). (We give a detailed sketch of our proof of Theorem 1.1 in Section 3.)

Importantly, our techniques apply under arbitrary convex and compact strategy spaces, thereby allowing for a direct extension to, for example, *normal-form coarse correlated equilibria* in *extensive-form games (EFGs)* [Moulin and Vial, 1978]. Another compelling aspect of our result is that both players employ a *constant* learning rate, a feature which has been extensively motivated in prior works (*e.g.*, see [Bailey and Piliouras, 2019, Piliouras et al., 2021, Golowich et al., 2020a]). Besides the improved performance guarantees observed in practice under a time-invariant learning rate, it arguably induces a more natural behavior from an economic standpoint. Finally, we corroborate our theoretical findings through experiments on standard benchmark extensive-form games in Section 4.2.

## 1.2 Related Work

**Nash Equilibria in Bimatrix Games** Bimatrix games is one of the most fundamental and well-studied classes of games. In a celebrated series of works, it was shown that computing a NE in bimatrix games does not admit a fully polynomial-time approximation scheme (FPTAS), unless every problem in PPAD [Papadimitriou, 1994] can be solved in polynomial time [Chen et al., 2009, Daskalakis et al.,

---

[2]Closeness to NE is always implied in terms of the best response gap, not the distance to the set of NE.

[3]To explain this point further, it is important to connect *last-iterate convergence* with NE. Namely, if all players employ no-regret learning algorithms and the dynamics converge pointwise, then it is easy to see that the limit point has to be a NE. On the other hand, correlated equilibrium concepts are, by definition, incompatible with last-iterate convergence—at least under uncoupled dynamics. Indeed, correlation inherently *requires* cycling behavior. In light of this, there are games for which *stability can be at odds with efficiency* since NE can be dramatically more inefficient that correlation schemes [Moulin and Vial, 1978]. We also refer to our example in Section 4 where this phenomenon becomes clear; *c.f.*, see [Kleinberg et al., 2011].

2009]. In light of this intractability result, there has been a substantial amount of attention in deriving polynomial-time approximation algorithms for NE in bimatrix games [Daskalakis et al., 2006a, 2007, Bosse et al., 2010, Tsaknakis and Spirakis, 2008, Kontogiannis and Spirakis, 2010, Daskalakis and Papadimitriou, 2009b, Deligkas et al., 2022a,b]. In fact, even computing a sufficiently small *constant* approximation essentially requires quasi-polynomial time [Rubinstein, 2016, Kothari and Mehta, 2018], subject to the exponential-time hypothesis for PPAD [Babichenko et al., 2016], matching the seminal QPTAS of Lipton et al. [2003]. It is also worth stressing that, in terms of computing Nash equilibria, there is a polynomial-time reduction from any game with a *succinct representation* to a bimatrix game [Daskalakis et al., 2006b]; this illustrates the generality of bimatrix games.

**Near-Optimal Regret in Games**  In a pioneering work by Daskalakis et al. [2011], it was shown that there exist uncoupled no-regret learning dynamics such that when employed by *both* players in a *zero-sum* game, the cumulative regret incurred by each player is bounded by $O(\log T)$, substantially improving over the $\Omega(\sqrt{T})$ lower bound under *adversarial utilities*. Thereafter, there has been a considerable amount of effort in simplifying and extending the previous results along several lines [Rakhlin and Sridharan, 2013a, Syrgkanis et al., 2015, Chen and Peng, 2020, Foster et al., 2016, Wei and Luo, 2018, Farina et al., 2019a, Daskalakis and Golowich, 2022, Daskalakis et al., 2021, Piliouras et al., 2021]. This line of work was culminated in a recent result due to Daskalakis et al. [2021], establishing $O(\text{polylog}(T))$ individual regret when all players employ an *optimistic* variant of *multiplicative weights update* in general games, nearly-matching the lower bound in [Daskalakis et al., 2011]. Nevertheless, the optimality here is only *existential*: there *exist* games—in fact, zero-sum—for which the guarantee in [Daskalakis et al., 2021] is essentially unimprovable. But we argue that this is not a good enough justification for considering the problem of no-regret learning in games settled. As we show in this paper, substantially more refined guarantees are possible beyond zero-sum games.

**Last-Iterate Convergence**  A folklore phenomenon in the theory of learning in games is that broad families of no-regret algorithms exhibit *cyclic* or even *chaotic behavior* [Sato et al., 2002, Sandholm, 2010, Mertikopoulos et al., 2018, Andrade et al., 2021, Cheung and Piliouras, 2020, Cheung and Tao, 2021], even in low-dimensional zero-sum games. A compelling approach to ameliorate this problem was proposed by Daskalakis et al. [2018], showing that an *optimistic* variant of gradient descent guarantees last-iterate convergence in unconstrained bilinear (zero-sum) games. Thereafter, there has been a tremendous amount of interest in strengthening and extending their result in different regimes; for a highly incomplete list, we refer to [Daskalakis and Panageas, 2019, 2018, Wei et al., 2021a, Golowich et al., 2020a,b, Hsieh et al., 2021, Zhou et al., 2017, Lin et al., 2020, Mertikopoulos et al., 2019, Liang and Stokes, 2019, Zhang and Yu, 2020, Mokhtari et al., 2020, Daskalakis et al., 2020, Wei et al., 2021b, Anagnostides and Panageas, 2021], and references therein. The precursor of our main result is the recent paper of Anagnostides et al. [2022a] that obtained similar results but for the *sum* of the players' regrets, which does not have implications in terms of convergence to CCE.

## 2  Preliminaries

**Notation and Conventions**  We let $\mathbb{N} := \{1, 2, \dots\}$ be the set of natural numbers. The (discrete) time index is represented exclusively via the variable $t$. We also use the shorthand notation $[\![p]\!] := \{1, 2, \dots, p\}$. We let $\|\cdot\|$ be a norm on $\mathbb{R}^d$, for some $d \in \mathbb{N}$, and $\|\cdot\|_*$ be the *dual norm* of $\|\cdot\|$; namely, $\|\boldsymbol{v}\|_* := \sup_{\|\boldsymbol{u}\| \leq 1} \langle \boldsymbol{u}, \boldsymbol{v} \rangle$. For a (nonempty) convex and compact set $\mathcal{X} \subseteq \mathbb{R}^d$, we denote by $\Omega_{\mathcal{X}} := \max_{\boldsymbol{x}, \widehat{\boldsymbol{x}} \in \mathcal{X}} \|\boldsymbol{x} - \widehat{\boldsymbol{x}}\|$ the *diameter* of $\mathcal{X}$ with respect to $\|\cdot\|$; to lighten the notation, the underlying norm will be implicit in our notation. Moreover, we let $\|\mathcal{X}\| := \max_{\boldsymbol{x} \in \mathcal{X}} \|\boldsymbol{x}\|$. For a matrix $\mathbf{A}$ we let $\|\mathbf{A}\|_{\text{op}}$ be its *operator norm*: $\|\mathbf{A}\|_{\text{op}} := \sup_{\|\boldsymbol{u}\| \leq 1} \|\mathbf{A}\boldsymbol{u}\|_*$. We point out that $\|\mathbf{A}\|_{\text{op}} \neq 0$ when $\mathbf{A} \neq \mathbf{0}$. Finally, for a finite nonempty set $\mathcal{A}$, we let $\Delta(\mathcal{A}) := \left\{ \boldsymbol{x} \in \mathbb{R}^{|\mathcal{A}|}_{\geq 0} : \sum_{a \in \mathcal{A}} \boldsymbol{x}(a) = 1 \right\}$ be the *probability simplex* over $\mathcal{A}$, where $\boldsymbol{x}(a)$ is the coordinate of $\boldsymbol{x}$ corresponding to $a \in \mathcal{A}$.

### 2.1  Online Learning and Regret

Let $\mathcal{X}$ be a nonempty, convex and compact subset of $\mathbb{R}^d$, for some $d \in \mathbb{N}$. In the *online learning framework* the *learner* commits to a *strategy* $\boldsymbol{x}^{(t)} \in \mathcal{X}$ at every iteration $t \in \mathbb{N}$. Then, the learner receives as feedback from the environment a (linear) *utility function* $u^{(t)} : \mathcal{X} \ni \boldsymbol{x} \mapsto \langle \boldsymbol{x}, \boldsymbol{u}^{(t)} \rangle$, with $\boldsymbol{u}^{(t)} \in \mathbb{R}^d$, so that the utility received at time $t$ is given by $\langle \boldsymbol{x}^{(t)}, \boldsymbol{u}^{(t)} \rangle$. Based on that feedback, the

learner may adapt its next strategy. The framework is *online* in the sense that no information about future utilities is available. The canonical objective in this framework is to minimize the *cumulative external regret* (or simply *regret*), defined for a *time horizon* $T \in \mathbb{N}$ as follows.

$$\text{Reg}^T \coloneqq \max_{\boldsymbol{x}^* \in \mathcal{X}} \left\{ \sum_{t=1}^{T} \langle \boldsymbol{x}^*, \boldsymbol{u}^{(t)} \rangle \right\} - \sum_{t=1}^{T} \langle \boldsymbol{x}^{(t)}, \boldsymbol{u}^{(t)} \rangle, \tag{1}$$

That is, the performance is measured in terms of the optimal *fixed strategy in hindsight*.

## 2.2 Optimistic Regret Minimization

To leverage the additional structure in more benign environments, several *predictive* algorithms have been recently introduced [Hazan and Kale, 2011, Chiang et al., 2012, 2013, Rakhlin and Sridharan, 2013b,a, Syrgkanis et al., 2015, Farina et al., 2021]. In this paper we employ a predictive variant of MD, known as *optimistic mirror descent* [Chiang et al., 2012, Rakhlin and Sridharan, 2013a].

**Optimistic Mirror Descent**  Let $\mathcal{R}$ be a 1-strongly convex continuously differentiable regularizer (or *distance generating function (DGF)*) with respect to a norm $\| \cdot \|$ on $\mathbb{R}^d$. We say that $\mathcal{R}$ is *G-smooth*, with $G > 0$, if for any $\boldsymbol{x}, \widehat{\boldsymbol{x}} \in \mathcal{X}$,

$$\|\nabla \mathcal{R}(\boldsymbol{x}) - \nabla \mathcal{R}(\widehat{\boldsymbol{x}})\|_* \leq G \|\boldsymbol{x} - \widehat{\boldsymbol{x}}\|. \tag{2}$$

For example, the *Euclidean regularizer* $\mathcal{R}(\boldsymbol{x}) \coloneqq \frac{1}{2}\|\boldsymbol{x}\|_2^2$, which is 1-strongly convex with respect to the Euclidean norm $\| \cdot \|_2$, trivially satisfies the smoothness condition of (2) with $G = 1$; we recall that $\| \cdot \|_2$ is self-dual. Moreover, we let $D_{\mathcal{R}}(\boldsymbol{x} \parallel \widehat{\boldsymbol{x}}) \coloneqq \mathcal{R}(\boldsymbol{x}) - \mathcal{R}(\widehat{\boldsymbol{x}}) - \langle \nabla \mathcal{R}(\widehat{\boldsymbol{x}}), \boldsymbol{x} - \widehat{\boldsymbol{x}} \rangle$ be the *Bregman divergence* induced by $\mathcal{R}$ [Rockafellar, 1970]. *Optimistic mirror descent (OMD)*[4] is parameterized by a (dynamic) *prediction vector* $\boldsymbol{m}^{(t)} \in \mathbb{R}^d$, for every time $t \in \mathbb{N}$, and a *learning rate* $\eta > 0$, so that its update rule takes the following form for $t \in \mathbb{N}$:

$$\begin{aligned} \boldsymbol{x}^{(t)} &\coloneqq \arg\max_{\boldsymbol{x} \in \mathcal{X}} \left\{ \langle \boldsymbol{x}, \boldsymbol{m}^{(t)} \rangle - \frac{1}{\eta} D_{\mathcal{R}}(\boldsymbol{x} \parallel \widehat{\boldsymbol{x}}^{(t-1)}) \right\}; \\ \widehat{\boldsymbol{x}}^{(t)} &\coloneqq \arg\max_{\widehat{\boldsymbol{x}} \in \mathcal{X}} \left\{ \langle \widehat{\boldsymbol{x}}, \boldsymbol{u}^{(t)} \rangle - \frac{1}{\eta} D_{\mathcal{R}}(\widehat{\boldsymbol{x}} \parallel \widehat{\boldsymbol{x}}^{(t-1)}) \right\}. \end{aligned} \tag{OMD}$$

Further, $\widehat{\boldsymbol{x}}^{(0)} \coloneqq \arg\min_{\widehat{\boldsymbol{x}} \in \mathcal{X}} \mathcal{R}(\widehat{\boldsymbol{x}})$; for convenience, we also let $\boldsymbol{x}^{(0)} \coloneqq \widehat{\boldsymbol{x}}^{(0)}$. We will refer to $\widehat{\boldsymbol{x}}^{(0)}, \widehat{\boldsymbol{x}}^{(1)}, \dots$ as the *secondary* sequence of OMD, while $\boldsymbol{x}^{(0)}, \boldsymbol{x}^{(1)}, \dots$ is the *primary* sequence. We also let $\Omega_{\mathcal{R}} \coloneqq \max_{\boldsymbol{x} \in \mathcal{X}} D_{\mathcal{R}}(\boldsymbol{x} \parallel \widehat{\boldsymbol{x}}^{(0)})$ denote the $\mathcal{R}$-diameter of $\mathcal{X}$. As usual, we consider the "one-recency bias" prediction mechanism [Syrgkanis et al., 2015], wherein $\boldsymbol{m}^{(t)} \coloneqq \boldsymbol{u}^{(t-1)}$ for $t \in \mathbb{N}$. (To make $\boldsymbol{u}^{(0)}$ well-defined in games, each player initially receives the utility corresponding to the players' strategies at $t = 0$; this is only made for convenience in the analysis.)

An important special case of (OMD) arises under the Euclidean regularizer $\mathcal{R}(\boldsymbol{x}) = \frac{1}{2}\|\boldsymbol{x}\|_2^2$, in which case we refer to the dynamics as *optimistic gradient descent (OGD)*:

$$\begin{aligned} \boldsymbol{x}^{(t)} &\coloneqq \Pi_{\mathcal{X}} \left( \widehat{\boldsymbol{x}}^{(t-1)} + \eta \boldsymbol{m}^{(t)} \right); \\ \widehat{\boldsymbol{x}}^{(t)} &\coloneqq \Pi_{\mathcal{X}} \left( \widehat{\boldsymbol{x}}^{(t-1)} + \eta \boldsymbol{u}^{(t)} \right). \end{aligned} \tag{OGD}$$

Here, $\Pi_{\mathcal{X}}(\cdot)$ stands for the Euclidean projection to the set $\mathcal{X}$.

A fundamental property crystallized in [Syrgkanis et al., 2015], building on [Rakhlin and Sridharan, 2013a], is the *regret bounded by variation in utilities* (RVU). For our purposes, we will employ a refinement of the RVU bound for OMD, which follows from [Rakhlin and Sridharan, 2013a, Lemma 1 in the Full Version]—this can be extracted en route to the proof of [Syrgkanis et al., 2015, Theorem 18].

**Proposition 2.1** ([Rakhlin and Sridharan, 2013a, Syrgkanis et al., 2015]). *The regret of* (OMD) *with learning rate $\eta > 0$ can be bounded as*

$$\text{Reg}^T \leq \frac{\Omega_{\mathcal{R}}}{\eta} + \eta \sum_{t=1}^{T} \|\boldsymbol{u}^{(t)} - \boldsymbol{u}^{(t-1)}\|_*^2 - \frac{1}{4\eta} \sum_{t=1}^{T} \left( \|\boldsymbol{x}^{(t)} - \widehat{\boldsymbol{x}}^{(t)}\|^2 + \|\boldsymbol{x}^{(t)} - \widehat{\boldsymbol{x}}^{(t-1)}\|^2 \right).$$

---

[4]To avoid any confusion, we point out that OMD sometimes stands for *online* mirror descent in the literature.

For convenience, we will use a shorthand notation for the *second-order path length*:

$$\Sigma_{\mathcal{X}}^T := \sum_{t=1}^{T} \left( \|\boldsymbol{x}^{(t)} - \widehat{\boldsymbol{x}}^{(t)}\|^2 + \|\boldsymbol{x}^{(t)} - \widehat{\boldsymbol{x}}^{(t-1)}\|^2 \right). \tag{3}$$

### 2.3 No-Regret Learning and Coarse Correlated Equilibria

A folklore connection ensures that when all players in a general game employ a no-regret learning algorithm, the average correlated distribution of play converges to a *coarse correlated equilibrium (CCE)* [Aumann, 1974, Moulin and Vial, 1978]. Formally, let us first introduce the notion of a CCE in general *normal-form games (NFGs)*. To this end, we consider a set of $p$ players $[\![p]\!]$, with each player $i$ having a set of available actions $\mathcal{A}_i$. The *utility* of player $i \in [\![p]\!]$ is a function $u_i : \bigtimes_{i=1}^{p} \mathcal{A}_i \ni \boldsymbol{a} \mapsto \mathbb{R}$ indicating the utility received by player $i$ under the action profile $\boldsymbol{a} = (a_1, \ldots, a_p)$.

**Definition 2.2** (Approximate Coarse Correlated Equilibrium). A distribution $\boldsymbol{\mu}$ over the set $\bigtimes_{i=1}^{p} \mathcal{A}_i$ is an $\epsilon$-*approximate coarse correlated equilibrium*, with $\epsilon \geq 0$, if for any player $i \in [\![p]\!]$ and any unilateral deviation $a_i' \in \mathcal{A}_i$,

$$\mathbb{E}_{\boldsymbol{a} \sim \boldsymbol{\mu}} [u_i(a_i', \boldsymbol{a}_{-i})] \leq \mathbb{E}_{\boldsymbol{a} \sim \boldsymbol{\mu}} [u_i(\boldsymbol{a})] + \epsilon.$$

We are now ready to state the fundamental theorem connecting no-regret learning with CCE; for completeness, we include the short proof in Appendix A.

**Theorem 2.3** (Folklore). *Suppose that every player $i \in [\![p]\!]$ employs a no-regret learning algorithm with regret $\mathrm{Reg}_i^T$ up to time $T \in \mathbb{N}$. Moreover, let $\boldsymbol{\mu}^{(t)} := \boldsymbol{x}_1^{(t)} \otimes \cdots \otimes \boldsymbol{x}_p^{(t)}$ be the correlated distribution of play at time $t \in [\![T]\!]$, and $\bar{\boldsymbol{\mu}} := \frac{1}{T} \sum_{t=1}^{T} \boldsymbol{\mu}^{(t)}$ be the average correlated distribution of play up to time $T$. Then,*

$$\mathbb{E}_{\boldsymbol{a} \sim \bar{\boldsymbol{\mu}}} [u_i(a_i', \boldsymbol{a}_{-i})] \leq \mathbb{E}_{\boldsymbol{a} \sim \bar{\boldsymbol{\mu}}} [u_i(\boldsymbol{a})] + \frac{1}{T} \max_{i \in [\![p]\!]} \mathrm{Reg}_i^T.$$

We also introduce a refinement of CCE which we refer to as *strong CCE*:

**Definition 2.4** (Strong Coarse Correlated Equilibrium). A probability distribution $\boldsymbol{\mu}$ over the set $\bigtimes_{i=1}^{p} \mathcal{A}_i$ is an $\epsilon$-*strong coarse correlated equilibrium*, with $\epsilon \geq 0$, if for any player $i \in [\![p]\!]$ and any unilateral deviation $a_i' \in \mathcal{A}_i$,

$$\mathbb{E}_{\boldsymbol{a} \sim \boldsymbol{\mu}} [u_i(a_i', \boldsymbol{a}_{-i})] \leq \mathbb{E}_{\boldsymbol{a} \sim \boldsymbol{\mu}} [u_i(\boldsymbol{a})] - \epsilon.$$

In a strong CCE any deviation from the mediator's recommendation is not only worse, but can decrease the player's utility by a significant amount. In that sense, a strong CCE can be a much more self-enforcing equilibrium outcome, and, as such, arguably more likely to occur. A strong CCE (Definition 2.4) can be thought of as a standard CCE (Definition 2.2) but with a "negative approximation". We include a discussion and an illustration of strong CCE in Section 4.1.

### 2.4 Bimatrix Games

A *bimatrix game* involves two players. Each player has a set of strategies $\mathcal{X} \subseteq \mathbb{R}^n$ and $\mathcal{Y} \subseteq \mathbb{R}^m$, respectively. The (expected) *payoffs* of each player under strategies $(\boldsymbol{x}, \boldsymbol{y}) \in \mathcal{X} \times \mathcal{Y}$ is given by the bilinear forms $\boldsymbol{x}^\top \mathbf{A} \boldsymbol{y}$ and $\boldsymbol{x}^\top \mathbf{B} \boldsymbol{y}$, respectively. Here, $\mathbf{A}, \mathbf{B} \in \mathbb{R}^{n \times m}$ are the payoff matrices of the game.[5] As an example, the special case where $\mathcal{X} = \Delta(\mathcal{A}_{\mathcal{X}})$ and $\mathcal{Y} = \Delta(\mathcal{A}_{\mathcal{Y}})$ corresponds to normal-form games, but our current formulation captures *extensive-form games* as well. By convention, we will refer to the two players as player $\mathcal{X}$ and player $\mathcal{Y}$ respectively. Furthermore, the underlying bimatrix game will be referred to as $(\mathbf{A}, \mathbf{B})$, without specifying the strategy sets.

**Definition 2.5** (Approximate Nash Equilibrium). A pair of strategies $(\boldsymbol{x}^*, \boldsymbol{y}^*) \in \mathcal{X} \times \mathcal{Y}$ is an $\epsilon$-*approximate Nash equilibrium* of $(\mathbf{A}, \mathbf{B})$, for $\epsilon \geq 0$, if for any $(\boldsymbol{x}, \boldsymbol{y}) \in \mathcal{X} \times \mathcal{Y}$,

$$\begin{aligned} \boldsymbol{x}^\top \mathbf{A} \boldsymbol{y}^* &\leq (\boldsymbol{x}^*)^\top \mathbf{A} \boldsymbol{y}^* + \epsilon; \\ (\boldsymbol{x}^*)^\top \mathbf{B} \boldsymbol{y} &\leq (\boldsymbol{x}^*)^\top \mathbf{B} \boldsymbol{y}^* + \epsilon. \end{aligned} \tag{4}$$

To make Definition 2.5 meaningful, some normalization has to be imposed on the utilities. Here, we will assume that $\max_{\boldsymbol{y} \in \mathcal{Y}} \|\mathbf{A} \boldsymbol{y}\|_* \leq 1$ and $\max_{\boldsymbol{x} \in \mathcal{X}} \|\mathbf{B}^\top \boldsymbol{x}\|_* \leq 1$.

---

[5] We assume that $\mathbf{A} \neq \mathbf{0}$ and $\mathbf{B} \neq \mathbf{0}$. In the contrary case Theorem 1.1 follows trivially.

## 3 Main Result

In this section we sketch the main ingredients required for the proof of Theorem 1.1; all the proofs are deferred to Appendix A. Moreover, a detailed version of Theorem 1.1 for normal-form games under Euclidean regularization is given in Corollary A.2. In the sequel, we assume that players $\mathcal{X}$ and $\mathcal{Y}$ employ regularizers $\mathcal{R}_\mathcal{X}$ and $\mathcal{R}_\mathcal{Y}$, respectively, so that the regret of each player enjoys an RVU bound with respect to the same pair of dual norms $(\|\cdot\|, \|\cdot\|_*)$. The first step is to cast the refined RVU bound of Proposition 2.1 for bimatrix games.

**Corollary 3.1.** *Suppose that both players employ* (OMD) *with learning rate $\eta > 0$. Then,*

$$\mathrm{Reg}_\mathcal{X}^T \leq \frac{\Omega_{\mathcal{R}_\mathcal{X}}}{\eta} + \eta\|\mathbf{A}\|_{\mathrm{op}}^2 \sum_{t=1}^T \|\boldsymbol{y}^{(t)} - \boldsymbol{y}^{(t-1)}\|^2 - \frac{1}{4\eta}\sum_{t=1}^T \left(\|\boldsymbol{x}^{(t)} - \widehat{\boldsymbol{x}}^{(t)}\|^2 + \|\boldsymbol{x}^{(t)} - \widehat{\boldsymbol{x}}^{(t-1)}\|^2\right);$$

$$\mathrm{Reg}_\mathcal{Y}^T \leq \frac{\Omega_{\mathcal{R}_\mathcal{Y}}}{\eta} + \eta\|\mathbf{B}\|_{\mathrm{op}}^2 \sum_{t=1}^T \|\boldsymbol{x}^{(t)} - \boldsymbol{x}^{(t-1)}\|^2 - \frac{1}{4\eta}\sum_{t=1}^T \left(\|\boldsymbol{y}^{(t)} - \widehat{\boldsymbol{y}}^{(t)}\|^2 + \|\boldsymbol{y}^{(t)} - \widehat{\boldsymbol{y}}^{(t-1)}\|^2\right).$$

The next critical step consists of showing that approximate fixed points of (OMD) under smooth regularization—in the sense of (2)—correspond to approximate Nash equilibria (recall Definition 2.5) of the underlying bimatrix game, as we formalize below.

**Proposition 3.2** (Approximate Fixed Points of OMD)**.** *Consider a bimatrix game $(\mathbf{A}, \mathbf{B})$, and suppose that both players employ* (OMD) *with learning rate $\eta > 0$ and a $G$-smooth regularizer. Then, if $\|\boldsymbol{x}^{(t)} - \widehat{\boldsymbol{x}}^{(t-1)}\|, \|\widehat{\boldsymbol{x}}^{(t)} - \boldsymbol{x}^{(t)}\| \leq \epsilon\eta$ and $\|\boldsymbol{y}^{(t)} - \widehat{\boldsymbol{y}}^{(t-1)}\|, \|\widehat{\boldsymbol{y}}^{(t)} - \boldsymbol{y}^{(t)}\| \leq \epsilon\eta$, the pair $(\boldsymbol{x}^{(t)}, \boldsymbol{y}^{(t)})$ is a $(2\epsilon G\max\{\Omega_\mathcal{X}, \Omega_\mathcal{Y}\} + \epsilon\eta)$-approximate Nash equilibrium of $(\mathbf{A}, \mathbf{B})$.*

Indeed, we show that when the iterates of (OMD) do not change by much (relatively to the learning rate), each player is approximately best responding to the observed utility. The following ingredient is where we rely on the two-player aspect of the game. On a high level, we show that when the strategies of one of the players change fast over time, the other player ought to be "moving" rapidly as well.

**Lemma 3.3.** *Suppose that both players in a bimatrix game $(\mathbf{A}, \mathbf{B})$ employ* (OMD) *with learning rate $\eta > 0$. Then, for any $T \in \mathbb{N}$,*

$$\sum_{t=1}^T \|\boldsymbol{y}^{(t)} - \boldsymbol{y}^{(t-1)}\| \geq \frac{1}{2\eta\|\mathcal{X}\|\|\mathbf{A}\|_{\mathrm{op}}}\sum_{t=1}^T \left(\|\boldsymbol{x}^{(t)} - \widehat{\boldsymbol{x}}^{(t-1)}\|^2 + \|\widehat{\boldsymbol{x}}^{(t)} - \boldsymbol{x}^{(t)}\|^2\right) - \frac{2}{\|\mathbf{A}\|_{\mathrm{op}}};$$

$$\sum_{t=1}^T \|\boldsymbol{x}^{(t)} - \boldsymbol{x}^{(t-1)}\| \geq \frac{1}{2\eta\|\mathcal{Y}\|\|\mathbf{B}\|_{\mathrm{op}}}\sum_{t=1}^T \left(\|\boldsymbol{y}^{(t)} - \widehat{\boldsymbol{y}}^{(t-1)}\|^2 + \|\widehat{\boldsymbol{y}}^{(t)} - \boldsymbol{y}^{(t)}\|^2\right) - \frac{2}{\|\mathbf{B}\|_{\mathrm{op}}}.$$

The intuition is that as long as one player is "moving" substantially faster than the other player, its utility will be monotonically increasing as the (repeated) game progresses. But this cannot occur for too long—by a potential argument—since the utility of each player is bounded. We are now ready to state the main technical theorem.

**Theorem 3.4** (Linear Decay of Regret; Full Version in Theorem A.1)**.** *Suppose that both players in a bimatrix game $(\mathbf{A}, \mathbf{B})$ employ* (OMD) *with smooth regularizer, learning rate $\eta = O(\epsilon^2)$ and $T = \Omega\left(\frac{1}{\epsilon^4\eta^2}\right)$, for a sufficiently small fixed $\epsilon > 0$. Then, if the dynamics do not reach an $O(\epsilon)$-approximate NE, then*

$$\max\{\mathrm{Reg}_\mathcal{X}^T, \mathrm{Reg}_\mathcal{Y}^T\} \leq -\Omega(\epsilon^4\eta T).$$

By virtue of Theorem 2.3 and Proposition 3.2, Theorem 3.4 immediately implies Theorem 1.1. An illustration of the linear decay of regret in a bimatrix game can be seen in Figure 3. Before we sketch the proof of Theorem 3.4, let us point out the following useful lemma.

**Lemma 3.5** (Stability of OMD)**.** *Suppose that both players employ* (OMD) *with learning rate $\eta > 0$. Then, for any $t \in \mathbb{N}$,*

$$\|\boldsymbol{x}^{(t)} - \boldsymbol{x}^{(t-1)}\| \leq 3\eta;$$
$$\|\boldsymbol{y}^{(t)} - \boldsymbol{y}^{(t-1)}\| \leq 3\eta.$$

*Sketch Proof of Theorem 3.4.* When the iterates of (OMD) are $\Omega(\epsilon)$ from being a Nash equilibrium, Proposition 3.2 implies that $\Sigma_{\mathcal{X}}^T + \Sigma_{\mathcal{Y}}^T = \Omega(\epsilon^2\eta^2 T)$, where we used the notation of (3). Now suppose that $\Sigma_{\mathcal{X}}^T \geq \Sigma_{\mathcal{Y}}^T$. Then, using Corollary 3.1 we get that $\mathrm{Reg}_{\mathcal{X}}^T = -\Omega(\epsilon^2\eta T)$. For the regret of player $\mathcal{Y}$, we first use Lemma 3.3 to obtain that $\Sigma_{\mathcal{Y}}^T = \Omega(\epsilon^4\eta^2 T)$. Finally, using Corollary 3.1 and Lemma 3.5 we can conclude that $\mathrm{Reg}_{\mathcal{Y}}^T \leq \frac{\Omega_{\mathcal{R}_{\mathcal{Y}}}}{\eta} - \Omega(\epsilon^4\eta T) = -\Omega(\epsilon^4\eta T)$. □

While the dichotomy of Theorem 1.1 is based on whether only a *single* iterate is an $\epsilon$-approximate Nash equilibrium, our techniques also directly give analogous guarantees depending on whether *most*—say 99%—of the iterates are $\epsilon$-approximate Nash equilibria, as we formalize in Corollary A.3.

*Remark* 3.6 (Extensive-Form Games). Theorem 1.1 has direct implications for *normal-form coarse correlated equilibria (NFCCE)* [Moulin and Vial, 1978] in extensive-form games using the *sequence-form strategy representation* [Romanovskii, 1962, von Stengel, 1996, Koller et al., 1996].

*Remark* 3.7 (Multiplayer Games). Theorem 1.1 does not extend to arbitrary games with $p \geq 3$. To see this, consider a 3-player game for which the utility of player 3 does not depend on the strategies of the other players. Then, the regret of player 3 will be strictly positive—as long as the initialization differs from the optimal strategy. So, even if the dynamics are far from Nash equilibria, the CCE gap will always be strictly positive. At a superficial level, this issue occurs in games where the strategic interactions form, in some sense, multiple "connected components". Nevertheless, characterizing the multiplayer games for which Theorem 1.1 holds is an interesting question for the future.

## 4 Experiments

In this section we provide experiments supporting our theoretical findings. We start by analyzing the CCE and the behavior of (OGD) in a simple bimatrix NFG in Section 4.1, while in Section 4.2 we experiment with several benchmark games used in the EFG-solving literature.

### 4.1 An Illustrative Example

First, we study the $3 \times 3$ bimatrix normal-form game $(\mathbf{A}, \mathbf{B})$, where

$$\mathbf{A} := \begin{bmatrix} 1 & 0 & 0 \\ -1 & 1 & 0 \\ 0 & 0 & 1 \end{bmatrix}; \quad \mathbf{B} := \begin{bmatrix} 0 & 1 & 0 \\ 0 & 0 & 1 \\ 1 & 0 & 0 \end{bmatrix}. \tag{5}$$

It can be shown that this bimatrix game has a unique Nash equilibrium $(\boldsymbol{x}^*, \boldsymbol{y}^*)$ such that $\boldsymbol{x}^* = (\frac{1}{3}, \frac{1}{3}, \frac{1}{3})$ and $\boldsymbol{y}^* = (\frac{1}{4}, \frac{1}{2}, \frac{1}{4})$ [Avis et al., 2010]. Moreover, $(\boldsymbol{x}^*, \boldsymbol{y}^*)$ secures a social welfare $\mathrm{SW}(\boldsymbol{x}^*, \boldsymbol{y}^*) := (\boldsymbol{x}^*)^\top \mathbf{A} \boldsymbol{y}^* + (\boldsymbol{x}^*)^\top \mathbf{B} \boldsymbol{y}^* = \frac{1}{4} + \frac{1}{3} \approx 0.5833$. On the other hand, it is easy to see that there exists an exact CCE $\boldsymbol{\mu}^*$ such that $\mathrm{SW}(\boldsymbol{\mu}^*) = 1$.

In fact, this social welfare is optimal even without any incentive-compatibility constraints. Furthermore, using a linear programming solver, we find that the *strongest* CCE (in the sense of Definition 2.4) has a parameter of roughly 0.2083. The entire landscape of CCE associated with the bimatrix game (5) is illustrated in Figure 2. The blue region corresponds to strong CCE, under which *both* players obtain a high utility. On the other hand, configurations for which one of the players receives low utility are not incentive compatible.

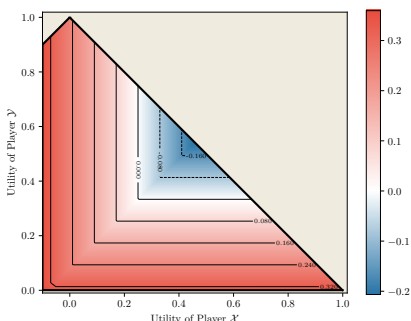

**Figure 2:** The maximum incentive-compatibility parameter of a CCE which guarantees a given pair of utilities.

Next, we focus on the behavior of the (OMD) dynamics. We let both players employ Euclidean regularization and learning rate $\eta := 0.1$. The convergence to CCE of the induced (OGD) dynamics is illustrated in Figure 3. In particular, after $T = 1000$ iterations the average

correlated distribution of play $\bar{\boldsymbol{\mu}}$ reads (with precision up to 4 decimal places)

$$\bar{\boldsymbol{\mu}} \approx \begin{bmatrix} 0.1594 & 0.1778 & 0.0048 \\ 0.0029 & 0.1614 & 0.1607 \\ 0.1642 & 0.0075 & 0.1613 \end{bmatrix}.$$

This correlated distribution secures a social welfare of $\mathrm{SW}(\bar{\boldsymbol{\mu}}) \approx 0.4793 + 0.5027 = 0.9819$. Thus, $\bar{\boldsymbol{\mu}}$ is near-optimal in terms of the obtained social welfare. As such, it substantially outperforms the efficiency of the Nash equilibrium $(\boldsymbol{x}^*, \boldsymbol{y}^*)$. Moreover, we see that $\bar{\boldsymbol{\mu}}$ is (approximately) a $0.1525$-strong CCE. Indeed, for player $\mathcal{X}$ the maximum possible utility attainable from a unilateral deviation is roughly $0.3268$, compared to $0.4793$ obtained under $\bar{\boldsymbol{\mu}}$; for player $\mathcal{Y}$ the maximum utility from a unilateral deviation is roughly $0.3420$, compared to $0.5027$. It is worth noting that (OGD) does *not* converge to the strongest possible CCE of the game, even under different random initializations. The results described here are robust to different initializations—although under the definition of (OGD) each player should start from the uniform distribution.

Finally, let us elaborate on Figure 1. The left image illustrates the Nash gap of the average strategies of (OGD) (with $\eta := 0.1$) in the zero-sum game $(\mathbf{A}, -\mathbf{A})$, while the right image shows the CCE gap of the average correlated distribution of play in the bimatrix game $(\mathbf{A}, \mathbf{B})$, as given in (5).

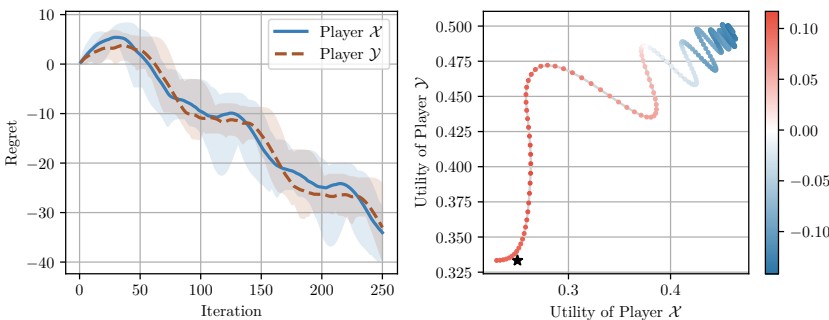

**Figure 3:** The convergence of (OGD) dynamics to CCE in the bimatrix game $(\mathbf{A}, \mathbf{B})$ introduced in (5). On the left image we plot the regret of each player under different random initializations; we recall that the maximum average regret is tantamount to the CCE gap. We see that after about 60 iterations both players experience *negative regret*—implying *exact* convergence to CCE. Furthermore, their regret decays linearly over time; this translates to convergence to strong CCE, as illustrated in the right image. More precisely, the color of each point (in the right image) corresponds to the CCE gap at the given iterate. Even when the initialization is "close" to the Nash equilibrium (represented by "*"), the dynamics manage to avoid it in search of more efficient outcomes. In fact, the limit CCE of the dynamics yields near-optimal social welfare.

## 4.2 Benchmark Games

Next, we illustrate the convergence of (OMD) on several benchmark bimatrix EFGs; namely: (i) *Sheriff* [Farina et al., 2019b]; (ii) *Liar's Dice* [Lisý et al., 2015]; (iii) *Battleship* [Farina et al., 2019b]; and (iv) *Goofspiel* [Ross, 1971]. A detailed description of the game instances we used for our experiments is included in Appendix B.

We instantiated (OMD) with Euclidean regularization. After a very mild tuning process, we chose for all games a (time-invariant) learning rate of $\eta = (2 \max\{\|\mathbf{A}\|_2, \|\mathbf{B}\|_2\})^{-1}$; here, $\|\cdot\|_2$ stands for the spectral norm of the corresponding matrix. For all games the initialization is chosen so that $\widehat{\boldsymbol{x}}^{(0)} := \arg\min_{\widehat{\boldsymbol{x}} \in \mathcal{X}} \mathcal{R}_{\mathcal{X}}(\widehat{\boldsymbol{x}})$ and $\widehat{\boldsymbol{y}}^{(0)} := \arg\min_{\widehat{\boldsymbol{y}} \in \mathcal{X}} \mathcal{R}_{\mathcal{Y}}(\widehat{\boldsymbol{y}})$, with the sole exception of Battleship for which that initialization is (virtually) already a Nash equilibrium. In light of this, for Battleship we initialized the dynamics in some arbitrary deterministic strategies; we stress that the conclusions derived here are robust to different initializations. Our results are summarized in Figure 4.

From these benchmark games, only Liar's Dice is constant-sum—in fact, zero-sum. Hence, as expected, the NE gap of the (OGD) dynamics essentially converges to $0$. Perhaps surprisingly, the same appears to hold for both Sheriff and Battleship. On the other hand, the dynamics exhibit a remarkably different behavior in Goofspiel. Indeed, although initially the dynamics appear to gradually converge to a Nash equilibrium, after about $600$ iterations the NE gap of the last iterate

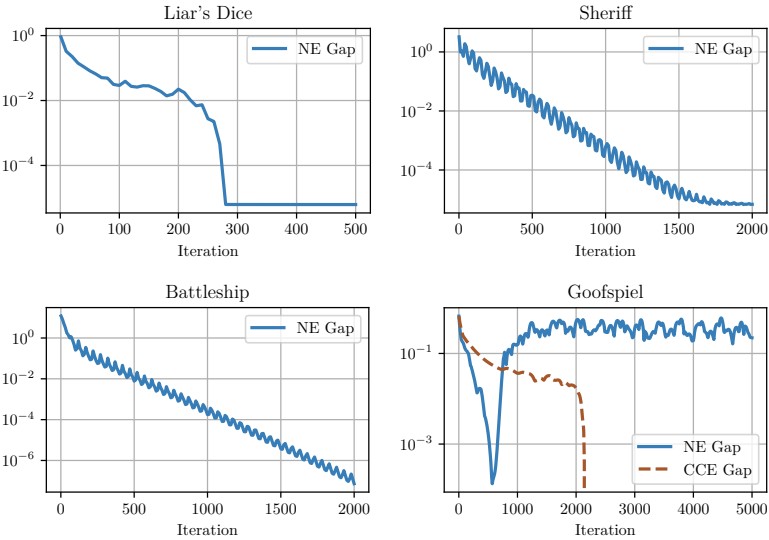

**Figure 4:** The NE gap of the last iterate and the CCE gap of the average correlated distribution of play under (OGD) in different benchmark games.

rapidly increases. Afterwards, the CCE gap starts to decay remarkably fast, eventually leading to a strong CCE. Specifically, after $T = 5000$ iterations the average correlated distribution of play is (roughly) a $0.0637$-strong CCE. These results are consistent with the predictions of Theorem 1.1.

## 5 Discussion

Our primary contribution was to establish a new characterization for the convergence properties of OMD—an uncoupled no-regret learning algorithm—when employed by both players in a general-sum game: OMD either reaches arbitrarily close to a Nash equilibrium, or, otherwise, *both* players experience $-\Omega(T)$ regret. Our results open several interesting avenues for future research:

- Can we extend Theorem 1.1 from coarse correlated equilibria to *correlated equilibria*? At the very least, such an extension would be particularly challenging since all the known *no-internal-regret* algorithms used for converging to correlated equilibria involve the stationary distribution of a Markov chain. While the recent reduction in [Anagnostides et al., 2022b] may seem useful, their technique only applies for (OMD) under entropic regularization. In contrast, our current argument crucially relies on the smoothness of the regularizer.

- For which classes of multiplayer games would Theorem 1.1 hold? As we pointed out in Remark 3.7, while Theorem 1.1 does not extend in arbitrary multiplayer games, it is still interesting to give sufficient conditions under which our results would carry over.

- Can we characterize the bimatrix games for which (OMD) exhibits last-iterate convergence? It should be noted that recent results seem to suggest that such a characterization may be too hard to obtain in general [Andrade et al., 2021].

- Finally, can we improve Theorem 1.1 in terms of the dependence on $T$ and $\eta$? For example, it would be interesting to extend Theorem 1.1 under a learning rate that does not depend on $\epsilon$. Some form of coupling in the spirit of *alternation* [Tammelin et al., 2015] could be useful in that direction.

## Acknowledgements

We are grateful to anonymous NeurIPS reviewers for many helpful comments. Ioannis Panageas is supported by a start-up grant. Part of this work was done while Ioannis Panageas was visiting the Simons Institute for the Theory of Computing. Tuomas Sandholm is supported by the National Science Foundation under grants IIS-1901403 and CCF-1733556.

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
