# A Omitted Proofs

In this section we include all the proofs deferred from the main body. For the convenience of the reader, all claims will be restated. Before we proceed with the proof of Theorem 2.3, let us point out some additional notational conventions. First, with a standard abuse of notation, we overload

$$u_i : \bigtimes_{i=1}^p \Delta(\mathcal{A}_i) \ni (\boldsymbol{x}_1, \ldots, \boldsymbol{x}_p) \mapsto \mathbb{E}_{\boldsymbol{a} \sim \boldsymbol{x}}[u_i(\boldsymbol{a})] = \sum_{\boldsymbol{a} \in \mathcal{A}} u_i(a_1, \ldots, a_p) \prod_{j \in [\![p]\!]} \boldsymbol{x}_j(a_j)$$

to denote the mixed extension of player's $i \in [\![p]\!]$ utility function, where $\mathcal{A} := \bigtimes_{j=1}^p \mathcal{A}_j$. Furthermore, for $a_i \in \mathcal{A}_i$ we let

$$u_i(a_i, \boldsymbol{x}_{-i}) := \sum_{\boldsymbol{a}_{-i} \in \mathcal{A}_{-i}} u_i(a_1, \ldots, a_i, \ldots, a_p) \prod_{j \neq i} \boldsymbol{x}_j(a_j),$$

where $\mathcal{A}_{-i} := \bigtimes_{j \neq i} \mathcal{A}_j$.

**Theorem 2.3** (Folklore). *Suppose that every player $i \in [\![p]\!]$ employs a no-regret learning algorithm with regret $\mathrm{Reg}_i^T$ up to time $T \in \mathbb{N}$. Moreover, let $\boldsymbol{\mu}^{(t)} := \boldsymbol{x}_1^{(t)} \otimes \cdots \otimes \boldsymbol{x}_p^{(t)}$ be the correlated distribution of play at time $t \in [\![T]\!]$, and $\bar{\boldsymbol{\mu}} := \frac{1}{T} \sum_{t=1}^T \boldsymbol{\mu}^{(t)}$ be the average correlated distribution of play up to time $T$. Then,*

$$\mathbb{E}_{\boldsymbol{a} \sim \bar{\boldsymbol{\mu}}}[u_i(a_i', \boldsymbol{a}_{-i})] \leq \mathbb{E}_{\boldsymbol{a} \sim \bar{\boldsymbol{\mu}}}[u_i(\boldsymbol{a})] + \frac{1}{T} \max_{i \in [\![p]\!]} \mathrm{Reg}_i^T .$$

*Proof.* By definition of regret (1), it follows that for any player $i \in [\![p]\!]$ and any possible deviation $a_i' \in \mathcal{A}_i$,

$$\sum_{t=1}^T u_i(a_i', \boldsymbol{x}_{-i}^{(t)}) - \sum_{t=1}^T u_i(\boldsymbol{x}^{(t)}) \leq \mathrm{Reg}_i^T . \tag{6}$$

Moreover,

$$\mathbb{E}_{\boldsymbol{a} \sim \bar{\boldsymbol{\mu}}}[u_i(\boldsymbol{a})] = \frac{1}{T} \sum_{t=1}^T \mathbb{E}_{\boldsymbol{a} \sim \boldsymbol{\mu}^{(t)}}[u_i(\boldsymbol{a})] = \frac{1}{T} \sum_{t=1}^T u_i(\boldsymbol{x}^{(t)});$$

and

$$\mathbb{E}_{\boldsymbol{a} \sim \bar{\boldsymbol{\mu}}}[u_i(a_i', \boldsymbol{a}_{-i})] = \frac{1}{T} \sum_{t=1}^T \mathbb{E}_{\boldsymbol{a} \sim \boldsymbol{\mu}^{(t)}}[u_i(a_i', \boldsymbol{a}_{-i})] = \frac{1}{T} \sum_{t=1}^T u_i(a_i', \boldsymbol{x}_{-i}^{(t)}).$$

Thus, the theorem follows directly from (6). $\qquad\square$

**Corollary 3.1.** *Suppose that both players employ* (OMD) *with learning rate $\eta > 0$. Then,*

$$\mathrm{Reg}_{\mathcal{X}}^T \leq \frac{\Omega_{\mathcal{R}_{\mathcal{X}}}}{\eta} + \eta \|\mathbf{A}\|_{\mathrm{op}}^2 \sum_{t=1}^T \|\boldsymbol{y}^{(t)} - \boldsymbol{y}^{(t-1)}\|^2 - \frac{1}{4\eta} \sum_{t=1}^T \left( \|\boldsymbol{x}^{(t)} - \widehat{\boldsymbol{x}}^{(t)}\|^2 + \|\boldsymbol{x}^{(t)} - \widehat{\boldsymbol{x}}^{(t-1)}\|^2 \right);$$

$$\mathrm{Reg}_{\mathcal{Y}}^T \leq \frac{\Omega_{\mathcal{R}_{\mathcal{Y}}}}{\eta} + \eta \|\mathbf{B}\|_{\mathrm{op}}^2 \sum_{t=1}^T \|\boldsymbol{x}^{(t)} - \boldsymbol{x}^{(t-1)}\|^2 - \frac{1}{4\eta} \sum_{t=1}^T \left( \|\boldsymbol{y}^{(t)} - \widehat{\boldsymbol{y}}^{(t)}\|^2 + \|\boldsymbol{y}^{(t)} - \widehat{\boldsymbol{y}}^{(t-1)}\|^2 \right).$$

*Proof.* First, the utility $\boldsymbol{u}_{\mathcal{X}}^{(t)}$ observed by player $\mathcal{X}$ at time $t \geq 0$ is equal to $\mathbf{A}\boldsymbol{y}^{(t)}$. Thus, the claimed bound for $\mathrm{Reg}_{\mathcal{X}}^T$ follows directly from Proposition 2.1 using the fact that $\|\mathbf{A}\boldsymbol{y}^{(t)} - \mathbf{A}\boldsymbol{y}^{(t-1)}\|_* \leq \|\mathbf{A}\|_{\mathrm{op}} \|\boldsymbol{y}^{(t)} - \boldsymbol{y}^{(t-1)}\|$. Similarly, the utility $\boldsymbol{u}_{\mathcal{Y}}^{(t)}$ observed by player $\mathcal{Y}$ at time $t \geq 0$ is tantamount to $\mathbf{B}^\top \boldsymbol{x}^{(t)}$. As a result, the bound for $\mathrm{Reg}_{\mathcal{Y}}^T$ follows from Proposition 2.1 and the fact that $\|\mathbf{B}^\top \boldsymbol{x}^{(t)} - \mathbf{B}^\top \boldsymbol{x}^{(t-1)}\|_* \leq \|\mathbf{B}\|_{\mathrm{op}} \|\boldsymbol{x}^{(t)} - \boldsymbol{x}^{(t-1)}\|$, since $\|\mathbf{B}^\top\|_{\mathrm{op}} = \|\mathbf{B}\|_{\mathrm{op}}$. $\qquad\square$

**Proposition 3.2** (Approximate Fixed Points of OMD). *Consider a bimatrix game $(\mathbf{A}, \mathbf{B})$, and suppose that both players employ* (OMD) *with learning rate $\eta > 0$ and a $G$-smooth regularizer. Then, if $\|\boldsymbol{x}^{(t)} - \widehat{\boldsymbol{x}}^{(t-1)}\|, \|\widehat{\boldsymbol{x}}^{(t)} - \boldsymbol{x}^{(t)}\| \leq \epsilon\eta$ and $\|\boldsymbol{y}^{(t)} - \widehat{\boldsymbol{y}}^{(t-1)}\|, \|\widehat{\boldsymbol{y}}^{(t)} - \boldsymbol{y}^{(t)}\| \leq \epsilon\eta$, the pair $(\boldsymbol{x}^{(t)}, \boldsymbol{y}^{(t)})$ is a $(2\epsilon G \max\{\Omega_{\mathcal{X}}, \Omega_{\mathcal{Y}}\} + \epsilon\eta)$-approximate Nash equilibrium of $(\mathbf{A}, \mathbf{B})$.*

*Proof.* First, by definition of the Bregman divergence, the update rule of (OMD) can be equivalently expressed as

$$\boldsymbol{x}^{(t)} := \arg\max_{\boldsymbol{x} \in \mathcal{X}} \left\{ \langle \boldsymbol{x}, \mathbf{A}\boldsymbol{y}^{(t-1)} \rangle - \frac{1}{\eta}\mathcal{R}(\boldsymbol{x}) + \frac{1}{\eta}\left\langle \boldsymbol{x}, \nabla\mathcal{R}(\widehat{\boldsymbol{x}}^{(t-1)}) \right\rangle \right\};$$

$$\widehat{\boldsymbol{x}}^{(t)} := \arg\max_{\widehat{\boldsymbol{x}} \in \mathcal{X}} \left\{ \langle \widehat{\boldsymbol{x}}, \mathbf{A}\boldsymbol{y}^{(t)} \rangle - \frac{1}{\eta}\mathcal{R}(\widehat{\boldsymbol{x}}) + \frac{1}{\eta}\left\langle \widehat{\boldsymbol{x}}, \nabla\mathcal{R}(\widehat{\boldsymbol{x}}^{(t-1)}) \right\rangle \right\}.$$

Now the maximization problem associated with the update rule of the secondary sequence can be equivalently cast in a variational inequality form (*e.g.*, see [Facchinei and Pang, 2003]):

$$\left\langle \widehat{\boldsymbol{x}} - \widehat{\boldsymbol{x}}^{(t)}, \mathbf{A}\boldsymbol{y}^{(t)} - \frac{1}{\eta}\left(\nabla\mathcal{R}(\widehat{\boldsymbol{x}}^{(t)}) - \nabla\mathcal{R}(\widehat{\boldsymbol{x}}^{(t-1)})\right) \right\rangle \leq 0, \quad \forall\widehat{\boldsymbol{x}} \in \mathcal{X}.$$

Thus,

$$\langle \widehat{\boldsymbol{x}} - \widehat{\boldsymbol{x}}^{(t)}, \mathbf{A}\boldsymbol{y}^{(t)} \rangle \leq \frac{1}{\eta}\left\langle \widehat{\boldsymbol{x}} - \widehat{\boldsymbol{x}}^{(t)}, \nabla\mathcal{R}(\widehat{\boldsymbol{x}}^{(t)}) - \nabla\mathcal{R}(\widehat{\boldsymbol{x}}^{(t-1)}) \right\rangle$$

$$\leq \frac{1}{\eta}\|\widehat{\boldsymbol{x}} - \widehat{\boldsymbol{x}}^{(t)}\|\|\nabla\mathcal{R}(\widehat{\boldsymbol{x}}^{(t)}) - \nabla\mathcal{R}(\widehat{\boldsymbol{x}}^{(t-1)})\|_* \tag{7}$$

$$\leq \frac{G}{\eta}\|\widehat{\boldsymbol{x}} - \widehat{\boldsymbol{x}}^{(t)}\|\|\widehat{\boldsymbol{x}}^{(t)} - \widehat{\boldsymbol{x}}^{(t-1)}\| \tag{8}$$

$$\leq 2\epsilon G\Omega_{\mathcal{X}}, \tag{9}$$

for any $\widehat{\boldsymbol{x}} \in \mathcal{X}$, where (7) derives from the Cauchy-Schwarz inequality; (8) follows since $\mathcal{R}_{\mathcal{X}}$ is assumed to be $G$-smooth; and (9) uses that $\|\widehat{\boldsymbol{x}}^{(t)} - \widehat{\boldsymbol{x}}^{(t-1)}\| \leq 2\epsilon\eta$, which in turn follows since $\|\widehat{\boldsymbol{x}}^{(t)} - \widehat{\boldsymbol{x}}^{(t-1)}\| \leq \|\widehat{\boldsymbol{x}}^{(t)} - \boldsymbol{x}^{(t)}\| + \|\boldsymbol{x}^{(t)} - \widehat{\boldsymbol{x}}^{(t-1)}\| \leq 2\epsilon\eta$ (triangle inequality), as well as the fact that, by definition, $\|\widehat{\boldsymbol{x}} - \widehat{\boldsymbol{x}}^{(t)}\| \leq \Omega_{\mathcal{X}}$ for any $\widehat{\boldsymbol{x}} \in \mathcal{X}$. As a result, we have shown that for any $\widehat{\boldsymbol{x}} \in \mathcal{X}$,

$$\langle \widehat{\boldsymbol{x}}^{(t)}, \mathbf{A}\boldsymbol{y}^{(t)} \rangle \geq \langle \widehat{\boldsymbol{x}}, \mathbf{A}\boldsymbol{y}^{(t)} \rangle - 2\epsilon G\Omega_{\mathcal{X}}. \tag{10}$$

Furthermore, by Cauchy-Schwarz inequality we have that

$$\langle \boldsymbol{x}^{(t)} - \widehat{\boldsymbol{x}}^{(t)}, \mathbf{A}\boldsymbol{y}^{(t)} \rangle \geq -\|\boldsymbol{x}^{(t)} - \widehat{\boldsymbol{x}}^{(t)}\|\|\mathbf{A}\boldsymbol{y}^{(t)}\|_* \geq -\epsilon\eta, \tag{11}$$

where we used the normalization assumption $\|\mathbf{A}\boldsymbol{y}^{(t)}\|_* \leq 1$. Thus, combing (11) with (10) yields that

$$\langle \boldsymbol{x}^{(t)}, \mathbf{A}\boldsymbol{y}^{(t)} \rangle \geq \langle \widehat{\boldsymbol{x}}^{(t)}, \mathbf{A}\boldsymbol{y}^{(t)} \rangle - \epsilon\eta \geq \langle \widehat{\boldsymbol{x}}, \mathbf{A}\boldsymbol{y}^{(t)} \rangle - 2\epsilon G\Omega_{\mathcal{X}} - \epsilon\eta, \tag{12}$$

for any $\widehat{\boldsymbol{x}} \in \mathcal{X}$. By symmetry, we analogously get that

$$\langle \boldsymbol{y}^{(t)}, \mathbf{B}^\top\boldsymbol{x}^{(t)} \rangle \geq \langle \widehat{\boldsymbol{y}}^{(t)}, \mathbf{B}^\top\boldsymbol{x}^{(t)} \rangle - \epsilon\eta \geq \langle \widehat{\boldsymbol{y}}, \mathbf{B}^\top\boldsymbol{x}^{(t)} \rangle - 2\epsilon G\Omega_{\mathcal{Y}} - \epsilon\eta, \tag{13}$$

for any $\widehat{\boldsymbol{y}} \in \mathcal{Y}$. Thus, recalling Definition 2.5, the claim follows from (12) and (13). $\square$

**Lemma 3.3.** *Suppose that both players in a bimatrix game* $(\mathbf{A}, \mathbf{B})$ *employ* (OMD) *with learning rate* $\eta > 0$*. Then, for any* $T \in \mathbb{N}$,

$$\sum_{t=1}^{T} \|\boldsymbol{y}^{(t)} - \boldsymbol{y}^{(t-1)}\| \geq \frac{1}{2\eta\|\mathcal{X}\|\|\mathbf{A}\|_{\mathrm{op}}} \sum_{t=1}^{T} \left( \|\boldsymbol{x}^{(t)} - \widehat{\boldsymbol{x}}^{(t-1)}\|^2 + \|\widehat{\boldsymbol{x}}^{(t)} - \boldsymbol{x}^{(t)}\|^2 \right) - \frac{2}{\|\mathbf{A}\|_{\mathrm{op}}};$$

$$\sum_{t=1}^{T} \|\boldsymbol{x}^{(t)} - \boldsymbol{x}^{(t-1)}\| \geq \frac{1}{2\eta\|\mathcal{Y}\|\|\mathbf{B}\|_{\mathrm{op}}} \sum_{t=1}^{T} \left( \|\boldsymbol{y}^{(t)} - \widehat{\boldsymbol{y}}^{(t-1)}\|^2 + \|\widehat{\boldsymbol{y}}^{(t)} - \boldsymbol{y}^{(t)}\|^2 \right) - \frac{2}{\|\mathbf{B}\|_{\mathrm{op}}}.$$

*Proof.* By 1-strong convexity of $\mathcal{R}_{\mathcal{X}}$ with respect to $\|\cdot\|$,

$$\langle \boldsymbol{x}^{(t)}, \mathbf{A}\boldsymbol{y}^{(t-1)} \rangle - \frac{1}{\eta}D_{\mathcal{R}_{\mathcal{X}}}(\boldsymbol{x}^{(t)} \parallel \widehat{\boldsymbol{x}}^{(t-1)}) - \langle \widehat{\boldsymbol{x}}^{(t-1)}, \mathbf{A}\boldsymbol{y}^{(t-1)} \rangle \geq \frac{1}{2\eta}\|\boldsymbol{x}^{(t)} - \widehat{\boldsymbol{x}}^{(t-1)}\|^2, \tag{14}$$

where we used the definition of the update rule of the primary sequence of (OMD). Similarly,

$$\langle \widehat{\boldsymbol{x}}^{(t)}, \mathbf{A}\boldsymbol{y}^{(t)} \rangle - \frac{1}{\eta}D_{\mathcal{R}_{\mathcal{X}}}(\widehat{\boldsymbol{x}}^{(t)} \parallel \widehat{\boldsymbol{x}}^{(t-1)}) - \langle \boldsymbol{x}^{(t)}, \mathbf{A}\boldsymbol{y}^{(t)} \rangle + \frac{1}{\eta}D_{\mathcal{R}_{\mathcal{X}}}(\boldsymbol{x}^{(t)} \parallel \widehat{\boldsymbol{x}}^{(t-1)}) \geq \frac{1}{2\eta}\|\widehat{\boldsymbol{x}}^{(t)} - \boldsymbol{x}^{(t)}\|^2. \tag{15}$$

Hence, summing (14) and (15) yields that

$$\langle \boldsymbol{x}^{(t)}, \mathbf{A}(\boldsymbol{y}^{(t-1)} - \boldsymbol{y}^{(t)}) \rangle \geq \frac{1}{2\eta}\left( \|\boldsymbol{x}^{(t)} - \widehat{\boldsymbol{x}}^{(t-1)}\|^2 + \|\widehat{\boldsymbol{x}}^{(t)} - \boldsymbol{x}^{(t)}\|^2 \right) - \langle \widehat{\boldsymbol{x}}^{(t)}, \mathbf{A}\boldsymbol{y}^{(t)} \rangle + \langle \widehat{\boldsymbol{x}}^{(t-1)}, \mathbf{A}\boldsymbol{y}^{(t-1)} \rangle,$$

where we used that $D_{\mathcal{R}_\mathcal{X}}(\widehat{\boldsymbol{x}}^{(t)} \parallel \widehat{\boldsymbol{x}}^{(t-1)}) \geq 0$. Thus, a telescopic summation over all $t \in [\![T]\!]$ implies that

$$\sum_{t=1}^{T} \langle \boldsymbol{x}^{(t)}, \mathbf{A}(\boldsymbol{y}^{(t-1)} - \boldsymbol{y}^{(t)}) \rangle \geq \frac{1}{2\eta} \sum_{t=1}^{T} \left( \|\boldsymbol{x}^{(t)} - \widehat{\boldsymbol{x}}^{(t-1)}\|^2 + \|\widehat{\boldsymbol{x}}^{(t)} - \boldsymbol{x}^{(t)}\|^2 \right) - 2\|\mathcal{X}\|, \quad (16)$$

since $-\langle \widehat{\boldsymbol{x}}^{(T)}, \mathbf{A}\boldsymbol{y}^{(T)} \rangle \geq -\|\widehat{\boldsymbol{x}}^{(T)}\|\|\mathbf{A}\boldsymbol{y}^{(T)}\|_* \geq -\|\mathcal{X}\|$ and $\langle \widehat{\boldsymbol{x}}^{(0)}, \mathbf{A}\boldsymbol{y}^{(0)} \rangle \geq -\|\widehat{\boldsymbol{x}}^{(0)}\|\|\mathbf{A}\boldsymbol{y}^{(0)}\|_* \geq -\|\mathcal{X}\|$, where we used the normalization assumption. Furthermore,

$$\langle \boldsymbol{x}^{(t)}, \mathbf{A}(\boldsymbol{y}^{(t-1)} - \boldsymbol{y}^{(t)}) \rangle \leq \|\boldsymbol{x}^{(t)}\|\|\mathbf{A}(\boldsymbol{y}^{(t)} - \boldsymbol{y}^{(t-1)})\|_* \leq \|\mathcal{X}\|\|\mathbf{A}\|_{\mathrm{op}}\|\boldsymbol{y}^{(t)} - \boldsymbol{y}^{(t-1)}\|.$$

Thus, combining this inequality with (16) implies that

$$\sum_{t=1}^{T} \|\boldsymbol{y}^{(t)} - \boldsymbol{y}^{(t-1)}\| \geq \frac{1}{2\eta\|\mathcal{X}\|\|\mathbf{A}\|_{\mathrm{op}}} \sum_{t=1}^{T} \left( \|\boldsymbol{x}^{(t)} - \widehat{\boldsymbol{x}}^{(t-1)}\|^2 + \|\widehat{\boldsymbol{x}}^{(t)} - \boldsymbol{x}^{(t)}\|^2 \right) - \frac{2}{\|\mathbf{A}\|_{\mathrm{op}}}.$$

This completes the first part of the claim. The second part follows analogously by symmetry. $\square$

**Lemma 3.5** (Stability of OMD). *Suppose that both players employ* (OMD) *with learning rate $\eta > 0$. Then, for any $t \in \mathbb{N}$,*

$$\|\boldsymbol{x}^{(t)} - \boldsymbol{x}^{(t-1)}\| \leq 3\eta;$$
$$\|\boldsymbol{y}^{(t)} - \boldsymbol{y}^{(t-1)}\| \leq 3\eta.$$

*Proof.* Fix any $t \in \mathbb{N}$. By definition of the primary sequence of (OMD),

$$\langle \boldsymbol{x}^{(t)}, \mathbf{A}\boldsymbol{y}^{(t-1)} \rangle - \frac{1}{\eta}D_\mathcal{R}(\boldsymbol{x}^{(t)} \parallel \widehat{\boldsymbol{x}}^{(t-1)}) - \langle \widehat{\boldsymbol{x}}^{(t-1)}, \mathbf{A}\boldsymbol{y}^{(t-1)} \rangle \geq \frac{1}{2\eta}\|\boldsymbol{x}^{(t)} - \widehat{\boldsymbol{x}}^{(t-1)}\|^2,$$

where we used the 1-strong convexity of the regularizer $\mathcal{R}_\mathcal{X}$ with respect to $\|\cdot\|$. In turn, this implies that

$$\langle \boldsymbol{x}^{(t)} - \widehat{\boldsymbol{x}}^{(t-1)}, \mathbf{A}\boldsymbol{y}^{(t-1)} \rangle \geq \frac{1}{\eta}\|\boldsymbol{x}^{(t)} - \widehat{\boldsymbol{x}}^{(t-1)}\|^2,$$

since $D_{\mathcal{R}_\mathcal{X}}(\boldsymbol{x}^{(t)} \parallel \widehat{\boldsymbol{x}}^{(t-1)}) \geq \frac{1}{2}\|\boldsymbol{x}^{(t)} - \widehat{\boldsymbol{x}}^{(t-1)}\|^2$ (by 1-strong convexity of $\mathcal{R}_\mathcal{X}$). Thus, an application of Cauchy-Schwarz inequality yields that

$$\|\boldsymbol{x}^{(t)} - \widehat{\boldsymbol{x}}^{(t-1)}\|^2 \leq \eta\|\boldsymbol{x}^{(t)} - \widehat{\boldsymbol{x}}^{(t-1)}\|\|\mathbf{A}\boldsymbol{y}^{(t-1)}\|_* \implies \|\boldsymbol{x}^{(t)} - \widehat{\boldsymbol{x}}^{(t-1)}\| \leq \eta, \quad (17)$$

since $\|\mathbf{A}\boldsymbol{y}^{(t-1)}\|_* \leq 1$ by the normalization assumption. Similar reasoning applied for the secondary sequence of (OMD) implies that for any $t \in \mathbb{N}$,

$$\|\widehat{\boldsymbol{x}}^{(t)} - \widehat{\boldsymbol{x}}^{(t-1)}\| \leq \eta. \quad (18)$$

Now if $t = 1$, it follows from (17) that $\|\boldsymbol{x}^{(t)} - \boldsymbol{x}^{(t-1)}\| = \|\boldsymbol{x}^{(t)} - \widehat{\boldsymbol{x}}^{(t-1)}\| \leq \eta$ since $\boldsymbol{x}^{(0)} = \widehat{\boldsymbol{x}}^{(0)}$. Otherwise, for $t \geq 2$, applying the triangle inequality yields that $\|\boldsymbol{x}^{(t)} - \boldsymbol{x}^{(t-1)}\| \leq \|\boldsymbol{x}^{(t)} - \widehat{\boldsymbol{x}}^{(t-1)}\| + \|\boldsymbol{x}^{(t-1)} - \widehat{\boldsymbol{x}}^{(t-2)}\| + \|\widehat{\boldsymbol{x}}^{(t-1)} - \widehat{\boldsymbol{x}}^{(t-2)}\| \leq 3\eta$ by (17) and (18). This completes the first part of the proof. Analogously, we conclude that $\|\boldsymbol{y}^{(t)} - \boldsymbol{y}^{(t-1)}\| \leq 3\eta$ for any $t \in \mathbb{N}$. $\square$

**Theorem A.1** (Linear Decay of Regret; Full Version of Theorem 3.4). *Suppose that both players in a bimatrix game $(\mathbf{A}, \mathbf{B})$ employ* (OMD) *with $G$-smooth regularizer, learning rate $\eta > 0$ such that*

$$\eta \leq \min\left\{ \frac{1}{4\max\{\|\mathbf{A}\|_{\mathrm{op}}, \|\mathbf{B}\|_{\mathrm{op}}\}}, \frac{\epsilon^2}{96\|\mathbf{A}\|_{\mathrm{op}}\|\mathbf{B}\|_{\mathrm{op}}\max\{\|\mathcal{X}\|, \|\mathcal{Y}\|\}} \right\}$$

*and*

$$T \geq \max\left\{ \frac{16\max\{\|\mathcal{X}\|, \|\mathcal{Y}\|\}}{\epsilon^2\eta}, \frac{32\max\{\Omega_{\mathcal{R}_\mathcal{X}}, \Omega_{\mathcal{R}_\mathcal{Y}}\}}{\epsilon^2\eta^2}, \frac{2048\max\{\Omega_{\mathcal{R}_\mathcal{Y}}\|\mathcal{X}\|^2\|\mathbf{A}\|_{\mathrm{op}}^2, \Omega_{\mathcal{R}_\mathcal{X}}\|\mathcal{Y}\|^2\|\mathbf{B}\|_{\mathrm{op}}^2\}}{\epsilon^4\eta^2} \right\},$$

*for some fixed $\epsilon > 0$. Then, if the dynamics do not reach a $(2\epsilon G\max\{\Omega_\mathcal{X}, \Omega_\mathcal{Y}\} + \epsilon\eta)$-approximate NE, then*

$$\max\{\mathrm{Reg}_\mathcal{X}^T, \mathrm{Reg}_\mathcal{Y}^T\} \leq -\min\left\{ \frac{\epsilon^2\eta}{32}, \frac{\epsilon^4\eta}{2048\max\{\|\mathcal{X}\|^2\|\mathbf{A}\|_{\mathrm{op}}^2, \|\mathcal{Y}\|^2\|\mathbf{B}\|_{\mathrm{op}}^2\}} \right\}T.$$

*Proof.* Suppose that there exists $t \in \llbracket T \rrbracket$ such that

$$\left( \|\boldsymbol{x}^{(t)} - \widehat{\boldsymbol{x}}^{(t)}\|^2 + \|\boldsymbol{x}^{(t)} - \widehat{\boldsymbol{x}}^{(t-1)}\|^2 \right) + \left( \|\boldsymbol{y}^{(t)} - \widehat{\boldsymbol{y}}^{(t)}\|^2 + \|\boldsymbol{y}^{(t)} - \widehat{\boldsymbol{y}}^{(t-1)}\|^2 \right) \leq \epsilon^2 \eta^2.$$

This would imply that $\|\boldsymbol{x}^{(t)} - \widehat{\boldsymbol{x}}^{(t)}\|, \|\boldsymbol{x}^{(t)} - \widehat{\boldsymbol{x}}^{(t-1)}\| \leq \epsilon\eta$ and $\|\boldsymbol{y}^{(t)} - \widehat{\boldsymbol{y}}^{(t)}\|, \|\boldsymbol{y}^{(t)} - \widehat{\boldsymbol{y}}^{(t-1)}\| \leq \epsilon\eta$. In turn, by Proposition 3.2 it follows that the pair of strategies $(\boldsymbol{x}^{(t)}, \boldsymbol{y}^{(t)})$ is a $(2\epsilon G \max\{\Omega_{\mathcal{X}}, \Omega_{\mathcal{Y}}\} + \epsilon\eta)$-approximate Nash equilibrium, contradicting our assumption. As a result, we conclude that for all $t \in \llbracket T \rrbracket$,

$$\left( \|\boldsymbol{x}^{(t)} - \widehat{\boldsymbol{x}}^{(t)}\|^2 + \|\boldsymbol{x}^{(t)} - \widehat{\boldsymbol{x}}^{(t-1)}\|^2 \right) + \left( \|\boldsymbol{y}^{(t)} - \widehat{\boldsymbol{y}}^{(t)}\|^2 + \|\boldsymbol{y}^{(t)} - \widehat{\boldsymbol{y}}^{(t-1)}\|^2 \right) \geq \epsilon^2 \eta^2.$$

Summing over all $t \in \llbracket T \rrbracket$ yields that

$$\sum_{t=1}^{T} \left( \|\boldsymbol{x}^{(t)} - \widehat{\boldsymbol{x}}^{(t)}\|^2 + \|\boldsymbol{x}^{(t)} - \widehat{\boldsymbol{x}}^{(t-1)}\|^2 \right) + \sum_{t=1}^{T} \left( \|\boldsymbol{y}^{(t)} - \widehat{\boldsymbol{y}}^{(t)}\|^2 + \|\boldsymbol{y}^{(t)} - \widehat{\boldsymbol{y}}^{(t-1)}\|^2 \right) \geq \epsilon^2 \eta^2 T. \tag{19}$$

We distinguish between two cases. First, we treat the case where

$$\sum_{t=1}^{T} \left( \|\boldsymbol{x}^{(t)} - \widehat{\boldsymbol{x}}^{(t)}\|^2 + \|\boldsymbol{x}^{(t)} - \widehat{\boldsymbol{x}}^{(t-1)}\|^2 \right) \geq \sum_{t=1}^{T} \left( \|\boldsymbol{y}^{(t)} - \widehat{\boldsymbol{y}}^{(t)}\|^2 + \|\boldsymbol{y}^{(t)} - \widehat{\boldsymbol{y}}^{(t-1)}\|^2 \right). \tag{20}$$

Then, by virtue of (19),

$$\sum_{t=1}^{T} \left( \|\boldsymbol{x}^{(t)} - \widehat{\boldsymbol{x}}^{(t)}\|^2 + \|\boldsymbol{x}^{(t)} - \widehat{\boldsymbol{x}}^{(t-1)}\|^2 \right) \geq \frac{\epsilon^2 \eta^2}{2} T. \tag{21}$$

Further, by the triangle inequality and Young's inequality,

$$\|\boldsymbol{y}^{(t)} - \boldsymbol{y}^{(t-1)}\|^2 \leq 2\|\boldsymbol{y}^{(t)} - \widehat{\boldsymbol{y}}^{(t-1)}\|^2 + 2\|\widehat{\boldsymbol{y}}^{(t-1)} - \boldsymbol{y}^{(t-1)}\|^2,$$

and summing over all $t \in \llbracket T \rrbracket$ yields that

$$\sum_{t=1}^{T} \|\boldsymbol{y}^{(t)} - \boldsymbol{y}^{(t-1)}\|^2 \leq 2\sum_{t=1}^{T} \|\boldsymbol{y}^{(t)} - \widehat{\boldsymbol{y}}^{(t-1)}\|^2 + 2\sum_{t=1}^{T} \|\widehat{\boldsymbol{y}}^{(t-1)} - \boldsymbol{y}^{(t-1)}\|^2$$

$$\leq 2\sum_{t=1}^{T} \|\boldsymbol{y}^{(t)} - \widehat{\boldsymbol{y}}^{(t-1)}\|^2 + 2\sum_{t=1}^{T} \|\widehat{\boldsymbol{y}}^{(t)} - \boldsymbol{y}^{(t)}\|^2,$$

where the last inequality follows since $\widehat{\boldsymbol{y}}^{(0)} = \boldsymbol{y}^{(0)}$. Hence, combining the latter bound with (20) implies that

$$\sum_{t=1}^{T} \left( \|\boldsymbol{x}^{(t)} - \widehat{\boldsymbol{x}}^{(t)}\|^2 + \|\boldsymbol{x}^{(t)} - \widehat{\boldsymbol{x}}^{(t-1)}\|^2 \right) \geq \frac{1}{2}\sum_{t=1}^{T} \|\boldsymbol{y}^{(t)} - \widehat{\boldsymbol{y}}^{(t-1)}\|^2. \tag{22}$$

Now we are ready to bound the regret of player $\mathcal{X}$. By Corollary 3.1,

$$\mathrm{Reg}_{\mathcal{X}}^{T} \leq \frac{\Omega_{\mathcal{R}_{\mathcal{X}}}}{\eta} + \eta\|\mathbf{A}\|_{\mathrm{op}}^2 \sum_{t=1}^{T} \|\boldsymbol{y}^{(t)} - \boldsymbol{y}^{(t-1)}\|^2 - \frac{1}{4\eta}\sum_{t=1}^{T} \left( \|\boldsymbol{x}^{(t)} - \widehat{\boldsymbol{x}}^{(t)}\|^2 + \|\boldsymbol{x}^{(t)} - \widehat{\boldsymbol{x}}^{(t-1)}\|^2 \right). \tag{23}$$

But (22) implies that

$$\eta\|\mathbf{A}\|_{\mathrm{op}}^2 \sum_{t=1}^{T} \|\boldsymbol{y}^{(t)} - \boldsymbol{y}^{(t-1)}\|^2 - \frac{1}{8\eta}\sum_{t=1}^{T} \left( \|\boldsymbol{x}^{(t)} - \widehat{\boldsymbol{x}}^{(t)}\|^2 + \|\boldsymbol{x}^{(t)} - \widehat{\boldsymbol{x}}^{(t-1)}\|^2 \right)$$

$$\leq \left( \eta\|\mathbf{A}\|_{\mathrm{op}}^2 - \frac{1}{16\eta} \right) \sum_{t=1}^{T} \|\boldsymbol{y}^{(t)} - \boldsymbol{y}^{(t-1)}\|^2 \leq 0,$$

since $\eta \leq \frac{1}{4\|\mathbf{A}\|_{\mathrm{op}}}$. From this we conclude that

$$\mathrm{Reg}_{\mathcal{X}}^T \leq \frac{\Omega_{\mathcal{R}_{\mathcal{X}}}}{\eta} - \frac{1}{8\eta}\sum_{t=1}^T \left( \|\boldsymbol{x}^{(t)} - \widehat{\boldsymbol{x}}^{(t)}\|^2 + \|\boldsymbol{x}^{(t)} - \widehat{\boldsymbol{x}}^{(t-1)}\|^2 \right) \leq \frac{\Omega_{\mathcal{R}_{\mathcal{X}}}}{\eta} - \frac{\epsilon^2\eta}{16}T \leq -\frac{\epsilon^2\eta}{32}T, \quad (24)$$

for $T \geq \frac{32\Omega_{\mathcal{R}_{\mathcal{X}}}}{\epsilon^2\eta^2}$, where we used (21). Next, we focus on the regret of player $\mathcal{Y}$. By (21) and Lemma 3.3,

$$\sum_{t=1}^T \|\boldsymbol{y}^{(t)} - \boldsymbol{y}^{(t-1)}\| \geq \frac{\epsilon^2\eta}{4\|\mathcal{X}\|\|\mathbf{A}\|_{\mathrm{op}}}T - \frac{2}{\|\mathbf{A}\|_{\mathrm{op}}} \geq \frac{\epsilon^2\eta}{8\|\mathcal{X}\|\|\mathbf{A}\|_{\mathrm{op}}}T, \quad (25)$$

since $T \geq \frac{16\|\mathcal{X}\|}{\epsilon^2\eta}$. Further, by Cauchy-Schwarz inequality,

$$\sum_{t=1}^T \|\boldsymbol{y}^{(t)} - \boldsymbol{y}^{(t-1)}\|^2 \geq \frac{1}{T}\left(\sum_{t=1}^T \|\boldsymbol{y}^{(t)} - \boldsymbol{y}^{(t-1)}\|\right)^2 \geq \frac{\epsilon^4\eta^2}{64\|\mathcal{X}\|^2\|\mathbf{A}\|_{\mathrm{op}}^2}T. \quad (26)$$

Now from Corollary 3.1, the regret of player $\mathcal{Y}$ can be bounded as

$$\mathrm{Reg}_{\mathcal{Y}}^T \leq \frac{\Omega_{\mathcal{R}_{\mathcal{Y}}}}{\eta} + \eta\|\mathbf{B}\|_{\mathrm{op}}^2 \sum_{t=1}^T \|\boldsymbol{x}^{(t)} - \boldsymbol{x}^{(t-1)}\|^2 - \frac{1}{4\eta}\sum_{t=1}^T\left(\|\boldsymbol{y}^{(t)} - \widehat{\boldsymbol{y}}^{(t)}\|^2 + \|\boldsymbol{y}^{(t)} - \widehat{\boldsymbol{y}}^{(t-1)}\|^2\right)$$

$$\leq \frac{\Omega_{\mathcal{R}_{\mathcal{Y}}}}{\eta} + \eta\|\mathbf{B}\|_{\mathrm{op}}^2 \sum_{t=1}^T \|\boldsymbol{x}^{(t)} - \boldsymbol{x}^{(t-1)}\|^2 - \frac{1}{8\eta}\sum_{t=1}^T \|\boldsymbol{y}^{(t)} - \boldsymbol{y}^{(t-1)}\|^2,$$

where we used that

$$\sum_{t=1}^T \|\boldsymbol{y}^{(t)} - \boldsymbol{y}^{(t-1)}\|^2 \leq 2\sum_{t=1}^T \|\boldsymbol{y}^{(t)} - \widehat{\boldsymbol{y}}^{(t-1)}\|^2 + 2\sum_{t=1}^T \|\widehat{\boldsymbol{y}}^{(t)} - \boldsymbol{y}^{(t)}\|^2.$$

Furthermore, by Lemma 3.5 and (26),

$$\eta\|\mathbf{B}\|_{\mathrm{op}}^2 \sum_{t=1}^T \|\boldsymbol{x}^{(t)} - \boldsymbol{x}^{(t-1)}\|^2 \leq 9\eta^3\|\mathbf{B}\|_{\mathrm{op}}^2 T \leq \frac{\epsilon^4\eta}{1024\|\mathcal{X}\|^2\|\mathbf{A}\|_{\mathrm{op}}^2}T \leq \frac{1}{16\eta}\sum_{t=1}^T \|\boldsymbol{y}^{(t)} - \boldsymbol{y}^{(t-1)}\|^2,$$

for $\eta \leq \epsilon^2(96\|\mathcal{X}\|\|\mathbf{A}\|_{\mathrm{op}}\|\mathbf{B}\|_{\mathrm{op}})^{-1}$. As a result, (26) implies that

$$\mathrm{Reg}_{\mathcal{Y}}^T \leq \frac{\Omega_{\mathcal{R}_{\mathcal{Y}}}}{\eta} - \frac{1}{16\eta}\sum_{t=1}^T \|\boldsymbol{y}^{(t)} - \boldsymbol{y}^{(t-1)}\|^2 \leq \frac{\Omega_{\mathcal{R}_{\mathcal{Y}}}}{\eta} - \frac{\epsilon^4\eta}{1024\|\mathcal{X}\|^2\|\mathbf{A}\|_{\mathrm{op}}^2}T \leq -\frac{\epsilon^4\eta}{2048\|\mathcal{X}\|^2\|\mathbf{A}\|_{\mathrm{op}}^2}T,$$

for $T \geq \frac{2048\Omega_{\mathcal{R}_{\mathcal{Y}}}\|\mathcal{X}\|^2\|\mathbf{A}\|_{\mathrm{op}}^2}{\epsilon^4\eta^2}$. Similarly, let us treat the case where

$$\sum_{t=1}^T\left(\|\boldsymbol{y}^{(t)} - \widehat{\boldsymbol{y}}^{(t)}\|^2 + \|\boldsymbol{y}^{(t)} - \widehat{\boldsymbol{y}}^{(t-1)}\|^2\right) \geq \sum_{t=1}^T\left(\|\boldsymbol{x}^{(t)} - \widehat{\boldsymbol{x}}^{(t)}\|^2 + \|\boldsymbol{x}^{(t)} - \widehat{\boldsymbol{x}}^{(t-1)}\|^2\right).$$

Then, for $\eta \leq \frac{1}{4\|\mathbf{B}\|_{\mathrm{op}}}$ and $T \geq \frac{32\Omega_{\mathcal{R}_{\mathcal{Y}}}}{\epsilon^2\eta^2}$,

$$\mathrm{Reg}_{\mathcal{Y}}^T \leq \frac{\Omega_{\mathcal{R}_{\mathcal{Y}}}}{\eta} - \frac{1}{8\eta}\sum_{t=1}^T\left(\|\boldsymbol{y}^{(t)} - \widehat{\boldsymbol{y}}^{(t)}\|^2 + \|\boldsymbol{y}^{(t)} - \widehat{\boldsymbol{y}}^{(t-1)}\|^2\right) \leq \frac{\Omega_{\mathcal{R}_{\mathcal{Y}}}}{\eta} - \frac{\epsilon^2\eta}{16}T \leq -\frac{\epsilon^2\eta}{32}T.$$

Moreover, for $T \geq \frac{16\|\mathcal{Y}\|}{\epsilon^2\eta}$,

$$\sum_{t=1}^T \|\boldsymbol{x}^{(t)} - \boldsymbol{x}^{(t-1)}\|^2 \geq \frac{\epsilon^4\eta^2}{64\|\mathcal{Y}\|^2\|\mathbf{B}\|_{\mathrm{op}}^2}T.$$

Thus, for $\eta \leq \epsilon^2(96\|\mathcal{Y}\|\|\mathbf{A}\|_{\mathrm{op}}\|\mathbf{B}\|_{\mathrm{op}})^{-1}$,

$$\eta\|\mathbf{A}\|_{\mathrm{op}}^2 \sum_{t=1}^T \|\boldsymbol{y}^{(t)} - \boldsymbol{y}^{(t-1)}\|^2 \leq 9\eta^3\|\mathbf{A}\|_{\mathrm{op}}^2 T \leq \frac{\epsilon^4\eta}{1024\|\mathcal{Y}\|^2\|\mathbf{B}\|_{\mathrm{op}}^2}T \leq \frac{1}{16\eta}\sum_{t=1}^T \|\boldsymbol{x}^{(t)} - \boldsymbol{x}^{(t-1)}\|^2.$$

Finally, for $T \geq \frac{2048 \Omega_{\mathcal{R}_{\mathcal{X}}} \|\mathcal{Y}\|^2 \|\mathbf{B}\|_{\mathrm{op}}^2}{\epsilon^4 \eta^2}$,

$$\mathrm{Reg}_{\mathcal{X}}^T \leq \frac{\Omega_{\mathcal{R}_{\mathcal{X}}}}{\eta} - \frac{1}{16\eta} \sum_{t=1}^{T} \|\boldsymbol{x}^{(t)} - \boldsymbol{x}^{(t-1)}\|^2 \leq \frac{\Omega_{\mathcal{R}_{\mathcal{X}}}}{\eta} - \frac{\epsilon^4 \eta}{1024 \|\mathcal{Y}\|^2 \|\mathbf{B}\|_{\mathrm{op}}^2} T \leq -\frac{\epsilon^4 \eta}{2048 \|\mathcal{Y}\|^2 \|\mathbf{B}\|_{\mathrm{op}}^2} T.$$

$\square$

Next, we state the implication of Theorem A.1 in normal-form games under (OGD). In that setting, it holds that $\|\mathcal{X}\|, \|\mathcal{Y}\| = 1$; $\Omega_{\mathcal{X}}, \Omega_{\mathcal{Y}} \leq \sqrt{2}$; $\Omega_{\mathcal{R}_{\mathcal{X}}}, \Omega_{\mathcal{R}_{\mathcal{Y}}} \leq 1$; and $G = 1$. Thus, we obtain the following simplified statement.

**Corollary A.2** (OGD in Normal-Form Games). *Suppose that both players in a bimatrix game* $(\mathbf{A}, \mathbf{B})$ *employ* (OGD) *with learning rate* $\eta > 0$ *such that*

$$\eta \leq \min \left\{ \frac{1}{4 \max\{\|\mathbf{A}\|_{\mathrm{op}}, \|\mathbf{B}\|_{\mathrm{op}}\}}, \frac{\epsilon^2}{96 \|\mathbf{A}\|_{\mathrm{op}} \|\mathbf{B}\|_{\mathrm{op}}} \right\}$$

*and*

$$T \geq \max \left\{ \frac{16}{\epsilon^2 \eta}, \frac{32}{\epsilon^2 \eta^2}, \frac{2048 \max\{\|\mathbf{A}\|_{\mathrm{op}}^2, \|\mathbf{B}\|_{\mathrm{op}}^2\}}{\epsilon^4 \eta^2} \right\},$$

*for some fixed* $\epsilon > 0$. *Then,*

- *Either there exists* $t \in \llbracket T \rrbracket$ *such that the pair of strategies* $(\boldsymbol{x}^{(t)}, \boldsymbol{y}^{(t)}) \in \mathcal{X} \times \mathcal{Y}$ *constitutes an* $\epsilon(3 + \eta)$-*approximate Nash equilibrium;*

- *Or, otherwise, the average correlated distribution of play after* $T$ *repetitions of the game is a*

$$\min \left\{ \frac{\epsilon^2 \eta}{32}, \frac{\epsilon^4 \eta}{2048 \max\{\|\mathbf{A}\|_{\mathrm{op}}^2, \|\mathbf{B}\|_{\mathrm{op}}^2\}} \right\} - \textit{strong coarse correlated equilibrium.}$$

Finally, we state an extension of Theorem 1.1 that establishes a dichotomy based on whether *most* of the iterates are approximate Nash equilibria—not just a *single* iterate. The proof is almost identical to the argument of Theorem A.1, and is therefore omitted.

**Corollary A.3.** *Suppose that both players in a bimatrix game employ* (OGD) *with learning rate* $\eta = O(\epsilon^2 \delta)$ *and* $T = \Omega\left(\frac{1}{\eta^2 \epsilon^4 \delta^2}\right)$ *repetitions, for a sufficiently small* $\epsilon > 0$ *and* $\delta \in (0, 1)$. *Then,*

- *Either a* $1 - \delta$ *fraction of the iterates is an* $\epsilon$-*approximate Nash equilibrium;*

- *Or, otherwise, the average correlated distribution of play is an* $\Omega(\epsilon^4 \eta \delta^2)$-*strong CCE.*

## B  Description of the Game Instances

In this section we provide a detailed description of the game instances we used in our experiments in Section 4.2.

**Liar's Dice**   The first game we experimented on is *Liar's dice*, a popular benchmark introduced by Lisý et al. [2015]. In our instantiation, each of the two players initially privately roles a *single* unbiased 4-face die. Then, the first player announces any face value up to 4, as well as the minimum number of dice the player believes have that value (among the dice of both players). Subsequently, each player in its own turn can either make a higher bid, or challenge the claim made by the previous player by declaring that player a "liar". In particular, a bid is higher than the previous one if either the face value is higher, or if the claimed number of dices is greater. In case the current player challenges the previous bid, all dice have to be revealed. If the claim was valid, the last bidder wins and receives a reward of $+1$, while the challenger incurs a negative payoff of $-1$. Otherwise, the utilities obtained are reversed.

**Sheriff**  Our second benchmark is a bargaining game inspired by the board game *Sheriff of Nottingham*, introduced by Farina et al. [2019b]. This game consists of two players: the *smuggler* and the *sheriff*. In our instantiation, the smuggler initially selects a number $n \in \{0, 1, 2, 3, 4, 5\}$ which corresponds to the number of *illegal items* to be loaded in the cargo. Each illegal item has a fixed value of 1. Next, 2 rounds of bargaining between the two players follow. At each round, the smuggler decides on a *bribe* ranging from 0 to $b := 3$ (inclusive), and the sheriff must decide whether or not the cargo will be inspected given the bribe amount. The sheriff's decision is binding only in the *last round* of bargaining: if the sheriff accepts the bribe, the game stops with the smuggler obtaining a utility of $n$ minus the bribe amount $b$ proposed in the last bargaining round, while the sheriff receives a utility equal to $b$. In contrast, if the sheriff does not accept the bribe in last bargaining round and decides to inspect the cargo, there are two possible alternatives:

- If the cargo has no illegal items (*i.e.* $n = 0$), the smuggler receives the fixed amount of 3, while sheriff incurs a negative payoff of $-3$;
- Otherwise, the utility of the smuggler is set to $-2n$, while the utility of the Sheriff is $2n$.

**Battleship**  Our next benchmark is *Battleship*, a parametric version of the popular board game introduced in [Farina et al., 2019b]. At the beginning, each player secretly places its ships on separate locations on a grid of size $2 \times 2$. Every ship has size 1 and a value of 4, and the placement is such that there is no overlap with any other ship. After the placement, players take turns at "firing" at their opponent's ships. The game proceeds until either one player has sunk all of the opponent's ships, or each player has completed $r = 2$ rounds of firing. At the end of the game, each player's payoff is the sum of the values of the opponent's ships that were sunk, minus the sum of the values of the ships that the player has lost *multiplied by two*. The latter modification makes the game general-sum, and incentivizes players to be more risk-averse.

**Goofspiel**  Our final benchmark is *Goofspiel*, introduced by Ross [1971]. In this game every player has a hand of cards numbered from 1 to $h$, where in our instantiation $h := 3$. An additional stack of $h$ cards is shuffled and singled out as winning the current prize. In every turn a prize card is revealed, and players privately choose one of their cards to bid. The player with the highest card wins the current prize, while in case of a tie the prize card is discarded. Due to this tie-breaking mechanism, even two-player instances are general-sum. After the completion of $h$ turns, players obtain the sum of the values of the prize cards they have won. Further, the instances we consider are of *limited information*—the actions of the other player are observed only at the end of the game. This makes the game strategically more involved as each player has less information about the opponent's actions.