# OpenReview forum: "Optimistic Mirror Descent Either Converges to Nash or to Strong Coarse Correlated Equilibria in Bimatrix Games"
_NeurIPS.cc/2022/Conference — NeurIPS 2022 Accept_

### Official Review · Reviewer_Yw3X · 2022-07-06

**Rating:** 5
**Confidence:** 4
**Soundness:** 3 good
**Presentation:** 3 good
**Contribution:** 3 good

**Summary:**

This paper studied the dynamic behaviour of Optimistic Mirror Descent beyond two-player zero-sum games and consider two-player general sum games (bimatrix games). It shows that the dynamic either visits an $\epsilon$-NE or achieve a strong coarse correlated equilibria.
In order to prove that, the authors observe the correlation in behaviour of two players 's strategies (Lemma 3.3 in Appendix) along with the existing regret bound of OMD (Proposition 2.1).

**Questions:**

I have a few questions to better understand the paper.

1. I am wondering whether the existence of a strong CCE is guaranteed in every bimatrix game?
2. In experiments, do the authors observe chaos behaviour of OMD? (e.g., the dynamic keeps approaching and moving away from a NE repeatedly)?



**Limitations:**

there is no negative societal impact.

**Strengths And Weaknesses:**

I really like the result of strong CCE and negative regret bound for both players (theorem 3.4) as they are new and fundamentally better compared to previous results. However, the condition in which the paper needs to achieve these results is hard to interpret. That is, the dynamic needs not to visit an $\epsilon$-Nash Equilibrium in the whole $T$ rounds. Hypothetically, OMD can have a chaotic behaviour while still visiting an $\epsilon$-Nash Equilibrium. This type of chaotic behaviour is known for many no-regret algorithms. Thus, the result of visiting $\epsilon$-Nash Equilibrium is less exciting and provides little information. Besides, visiting $\epsilon$-Nash Equilibrium and last round convergence are two very different, thus Theorem 1 does not imply the existing last round convergence result of OMD in zero-sum games.

The paper will be much stronger if the results in Theorem 1 can implement last-round convergence to NE (not visiting) or strong CCE. A relevant result in zero-sum games about $\epsilon$-NE leading to last round convergence can be found in "Last-Iterate Convergence: Zero-Sum Games and Constrained Min-Max Optimization".

---

> ### Author Response · Authors · 2022-07-29
> **Response to Reviewer Yw3X**
>
> We thank the reviewer for the helpful feedback. Below we address the main concerns.
>
> --- *“The result of visiting approximate Nash Equilibrium is less exciting and provides little information.”*
>
> Computing approximate Nash equilibria in bimatrix games is **one of the most fundamental problems in algorithmic game theory** with a tremendous amount of interest (please see our overview in Appendix A). Despite intense efforts, the best polynomial-time approximation is $\approx 1/3$, and for even smaller constant approximations the problem is known to require superpolynomial time (Rubinstein (2016)). So, even reaching an $\epsilon$-approximate Nash equilibrium, with a small constant $\epsilon$, is clearly remarkable, especially as it derives from efficient uncoupled no-regret learning dynamics, which are also scalable in very large games.
>
> --- *“The paper will be much stronger if the results in Theorem 1 can implement last-round convergence to NE (not visiting) or strong CCE. A relevant result in zero-sum games about $\epsilon$-NE leading to last round convergence can be found in Last-Iterate Convergence: Zero-Sum Games and Constrained Min-Max Optimization.”*
>
> There is a direct way of extending Theorem 1 so that **most of the strategies** (say a $1 - \delta$ fraction of them) are either $\epsilon$-approximate Nash equilibria, or we obtain a strong CCE, **for any constant** $\delta > 0$. This statement is significantly stronger and provides much more information than the one we included in our first version as it applies to, e.g., a $0.99 = 1 - \delta$ fraction of the strategies---instead of a *single* strategy. Hopefully it will address the reviewer’s concern.
>
> In proof, the difference is that if a $\delta$ fraction of the iterates are far from being approximate Nash equilibria, then the sum of the players' second-order path lengths is now $\Omega(\epsilon^2 \eta^2 T \times \delta)$;  that is, it still grows linearly in $T$, but is now multiplied by a factor $\delta$---recall our proof-sketch for Theorem 3.4. The rest of the proof is a matter of direct calculations. We will make sure to formally establish this in our revised version.
>
> We also point out that it is possible to just check the NE-gap---the maximum of the best response gaps---at every iteration, and effectively terminate the dynamics after a sufficient accuracy has been reached. This preserves all of the interesting algorithmic implications of our main result.
>
> Regarding comparison with the paper “Last-Iterate Convergence: Zero-Sum Games and Constrained Min-Max Optimization,” that paper only considers **zero-sum games**.  General-sum games behave completely differently than zero-sum games, so one should not compare our result in general-sum games with a guarantee that only applies to zero-sum games.
>
> --- *“I am wondering whether the existence of a strong CCE is guaranteed in every bimatrix game?”*
>
> That’s a very interesting question. $\epsilon$-strong CCE with $\epsilon > 0$ are **not** guaranteed to exist in general bimatrix games (but of course 0-CCEs always exist). In particular, they do not exist for zero-sum games, or more broadly for *strategically zero-sum games*—games that “behave” like zero-sum. More precisely, in zero-sum games all CCE are 0-strong; this follows directly from the well-known collapse of CCE to NE in zero-sum games. It is a plausible conjecture that strong CCE exist if and only if the game is not strategically zero-sum, although we have not pursued this direction.
>
> --- *“In experiments, do the authors observe chaos behaviour of OMD? (e.g., the dynamic keeps approaching and moving away from a NE repeatedly)?”*
>
> In some of our experiments OMD exhibits recurrent/cycling behavior (e.g., see our Goofspiel plot); this seems to be aligned with the findings of Piliouras and Cheung (2020) about the chaotic behavior of OMD in general-sum games. But, to answer the reviewer’s question, we did not observe the dynamics periodically approaching and moving away from NE.

---

> > ### Author Response · Authors · 2022-08-08
> > **Thank you for the feedback. Have we addressed the concerns?**
> >
> > We thank again the reviewer for the helpful feedback. Given that the discussion period is soon coming at an end, please let us know if we have adequately addressed the concerns raised, and if the reviewer has any further questions.

---

### Official Review · Reviewer_Vmdx · 2022-07-11

**Rating:** 7
**Confidence:** 3
**Soundness:** 3 good
**Presentation:** 4 excellent
**Contribution:** 3 good

**Summary:**

Precisely as per the title, the paper shows that optimistic mirror descent (OMD) either converges to epsilon-approximate Nash equilibria, or the average emprical distribution of correlated play converges to an approximate-"strong-coarse correlated equilibrium (eps-SCC). Here an epsilon-strong-coarse correlated equilibrium is parameterized by epsilon, which dictates the extent to which players see a decrease in utility from unilaterally deviating from the correlated distribution of play (where deviations are such that a player uses a single fixed strategy rather than following the correlated distribution's signals). An interesting consequence of the result lies in the fact that for constant epsilon, an eps-SCC is in fact an exact coarse correlated equilibrium, hence if the dynamic has not reached an eps-approximate, it must forcibly give rise to an exact coarse correlated equilibrium. Finally, the authors provide an in-depth analysis/visualization of the performance of OMW on a specific 3x3 game instance, as well as general performance on several game benchmarks

**Questions:**

In the experimental section you mention the possibility of different initializations for OGD. Are there games where different intializations to the same game can give rise to either of the guarantees of your result (eps-Nash or eps-strong CCE)?

Supposing that players do not have access to the representation of the game and infer utilities from potentially noisy oracles, are results robust to such a setting?


**Limitations:**

The main limitation that came to mind was the applicability of methods to games with more than 2 players, but this was adequately adressed in the paper as a potential thread of future research.

**Strengths And Weaknesses:**

Strengths:
-The paper is very clear and results are well-motivated. A large part of the clarity stems from the fact that results are simple to state with interesting consequences. Furthermore, the overall intuition behind main proofs is well-explained in spite of technique details being in the appendix. Furthermore, the results will be of interest to the general community at Neurips.

Weaknesses:
-The paper could have had more exposition regarding the setting where players do not have access to the full representation of the bimatrix game but rather infer knowledge of the game through oracles.
-A minor point really, but it would be interesting to map the trajectory of play for the game instance in Section 4, especially for different intializations as mentioned at the end of 4.1. This would be especially useful to visualize for the main Lemmas of section 3.

Minor point:
In the appendix you mention that the state of the art for approximate eps-Nash being the Tsaknakis and Spirakis 0.3393 + delta result. Recently there has been a paper Deligkas et al. (https://arxiv.org/pdf/2204.11525.pdf) which improves this to 1/3 + delta

---

> ### Author Response · Authors · 2022-07-29
> **Response to Reviewer Vmdx**
>
> We thank the reviewer for the helpful feedback. Below we address the main questions.
>
> --- *“The paper could have had more exposition regarding the setting where players do not have access to the full representation of the bimatrix game but rather infer knowledge of the game through oracles”*
>
> There could be a misunderstanding here: we do **not** assume that the players have a representation of the bimatrix game. Instead, the players initially have no knowledge about the game whatsoever, and they gradually elicit information about their utilities via utility oracles (full-information and noiseless). In particular, we operate within the standard uncoupled no-regret learning setting.
>
> --- *“Minor point: In the appendix you mention that the state of the art for approximate $epsilon$-Nash being the Tsaknakis and Spirakis $0.3393 + \delta$ result. Recently there has been a paper Deligkas et al.[...]”*
>
> We thank the reviewer for pointing this out; we were informed about that improvement just after the NeurIPS submission. We will include it in the revised version.
>
> --- *“In the experimental section you mention the possibility of different initializations for OGD. Are there games where different intializations to the same game can give rise to either of the guarantees of your result (eps-Nash or eps-strong CCE)?”*
>
> The answer is yes, at least in this sense: Nash equilibria are fixed points for OMD, so initializing at a Nash equilibrium will ensure that OMD will stay (and hence converge to) that Nash equilibrium. On the other hand, in our example in Section 4.1 we observed that the Nash equilibrium appears to be “repelling”---the plots that we illustrate also apply under random initializations. In other words, in that example it seems that we get strong CCE “for almost all initializations”---unless we start from the Nash equilibrium. But the reviewer’s point raises the very interesting question of whether either of the guarantees of our main theorem can arise under a non-trivial (e.g. nonzero) measure of initializations. We suspect that the answer is yes, but we have no concrete examples as of yet.
>
> --- *“Supposing that players do not have access to the representation of the game and infer utilities from potentially noisy oracles, are results robust to such a setting?”*
>
> Extending our results under noisy oracles is indeed a very interesting question—although we again note that we do not assume that players have access to the representation of the game. The answer to this question largely depends on the model of the noisy oracles. It is a plausible conjecture that under natural assumptions similar results can be derived, but we have not pursued this direction.
>
> One notable comment here is that extending our results to the bandit setting—where the player does not observe the entire utility vector, but only the utility that corresponds to the player’s action at that round—seems challenging. In particular, even obtaining a second-order path length bound for the regret (which is weaker property than the RVU bound) is a notorious open question in the literature on bandit learning.

---

### Official Review · Reviewer_6PC8 · 2022-07-11

**Rating:** 7
**Confidence:** 4
**Soundness:** 3 good
**Presentation:** 3 good
**Contribution:** 3 good

**Summary:**

This paper proves a new phenomenon about the Optimistic Mirror Descent (OMD) algorithm in two-player general-sum matrix games (bimatrix games): The iterates either converge to an approximate Nash Equilibrium (NE), or converge to a Strong Coarse Correlated Equilibrium (CCE). This result links and improves over two existing understandings: (1) Convergence to NE is unlikely to be generally achievable by any efficient algorithm due to its PPAD-hardness; (2) Convergence to approximate CCE is achievable by any no-regret algorithm, but it is unclear whether such CCE is a strong CCE (in the present paper’s sense).

**Questions:**

Please find some of my questions in the “weaknesses” part.

Additional questions:

How is Lemma 3.3 different from standard RVU bound (e.g. Proposition 2.1 & Corollary 3.1)? Skimming over the proof, it seems not a corollary of the RVU bound, but the proof steps are quite similar to that of RVU. Could the authors provide more intuition on this Lemma?

Last sentence in abstract, “our results suggest that cycling behavior of no-regret learning algorithms in games can be justified in terms of efficiency”--- is a bit unclear to me what it means? Does “efficiency” mean convergence to Strong CCE? Or better social welfare? Same goes with the statement on Line 54-55. Perhaps the authors could expand or modify these statements to clarify.


**Limitations:**

/

**Strengths And Weaknesses:**

Strengths:

The main message of this paper is conceptually quite interesting and new. Most existing works consider learning NE in two-player zero-sum games and CCE in multi-player general-sum games as parallel goals, as they can both be achieved by no-regret learning. On the other hand, learning NE in general-sum games is likely computationally challenging due to its PPAD hardness, and thus very sparsely studied in the context of no-regret learning (which mostly consider computationally efficient algorithms).

This paper essentially proves that, for the OMD algorithm and *two-player* general-sum games, if NE is not achieved, then convergence to CCE is stronger than standard bounds—you actually converge to $O(-\epsilon)$-CCE (what’s called the “strong CCE” in the paper) rather than $O(\epsilon)$-CCE. At a high level this seems to say something about the “optimization landscape” of bimatrix games, like it is perhaps more benign than the worst-case hardness / convergence results suggests.

Technically, it appears that the result follows by playing with the fundamental RVU property (and its proof) of OMD algorithms and making several smart observations. In particular, the proof (for the Reg_X part) follows by the observation that if NE is not achieved, then OMD iterates must move substantially (Proposition 3.2), which makes the negative term in the RVU dominate the positive term if learning rate is small enough, hence a *negative* regret and convergence to strong CCE. This argument appears new and may be applied in future work. It also gives me an impression that the RVU property may have other fruitful consequences yet to be discovered, despite its already powerful consequences for large bodies of work in this area.

It is also good to see the numerical experiments, in particular the fact that convergence to both NE and strong CCE are possible in general-sum games (so that the “either-or” statement in the theorem is unlikely to be improved, at least for this algorithm).


Weaknesses:

The result has a feeling of being slightly preliminary and not very complete. One caveat is the perhaps restrictive set of assumptions: global smoothness of regularizer, two-player games, and very small learning rate. In particular I am a bit bothered by the small learning rate, which is $O(\epsilon^2)$ compared with the standard $\Theta(1)$ learning rate for OMD type algorithms. This also makes convergence to NE/Strong CCE slower than standard $1/\epsilon^2$ time. I understand it is necessary in the current proof, but I wonder have the authors thought about larger learning rates (as mentioned in Appendix B)? Or alternatively, infinitesimally small learning rate like in gradient flow?

Also, the assumption of global smoothness of the regularizer rules out the very standard example of entropic regularizer on the probability simplex (whose smoothness grows to $\infty$ as we approach the boundary) for normal-form games, as mentioned in the open questions section in Appendix B. In numerical experiments, the authors consider L2 regularizers which are indeed globally smooth (thus satisfying the conditions of the theorem) but less standard.

Section 4.1 needs a bit more polishing. Some important quantities are not well-defined (e.g. what is “incentive-compatibility parameter?) Apart from that, Figure 2 is also a bit confusing (e.g. what are the policies used to plot Figure 2? Since policies have much more degrees of freedom than 2). Some of the numerical details in the text also does not seem too important and may be moved to the Appendix. I do like Figure 3 which shows very clearly the negative regret and the improvement in social welfare.

---

> ### Author Response · Authors · 2022-07-29
> **Response to Reviewer 6PC8**
>
> We thank the reviewer for the helpful comments. Below we address the main concerns.
>
> --- *“I am a bit bothered by the small learning rate [...]”*
>
> Our theorem allows one to pick $\epsilon$ to be a **universal constant** (for example, 0.1)---in which case $\eta = \Theta(1)$---while maintaining all of its interesting implications. Indeed, note that the best known polynomial-time algorithm only gives a $1/3$-approximate Nash equilibrium in bimatrix games; using any constant $\epsilon > 0$ smaller than $1/3$ would imply the best known polynomial-time algorithm for Nash equilibria—one of the most fundamental problems in algorithmic game theory. We will clarify this point further in the paper.
>
> We also remark that it is **not** typical to use $\eta = \Theta(1)$ in general-sum games; we are not aware of such guarantees (unlike in zero-sum games, where it is indeed standard to pick a constant $\eta = \Theta(1)$).
>
> --- *“Or alternatively, infinitesimally small learning rate like in gradient flow?”*
>
> Again, all of the interesting implication of our result hold even for constant learning rate $\eta = \Theta(1)$. General-sum games are very different from, e.g., zero-sum games, where one wants to select $\epsilon$ to be infinitesimally small in order to reach arbitrarily close to a Nash equilibrium. Our setting is fundamentally different for the reasons we described above—we do **not** require an infinitesimally small learning rate.
>
> --- *“I understand it is necessary in the current proof, but I wonder have the authors thought about larger learning rates [...]”*
>
> Notwithstanding the above remarks, this is an interesting question. We did pursue this direction extensively, but we now believe that it might be hard to do so—use a learning rate that does not depend on $\epsilon$---especially using uncoupled methods (as we do). It is plausible that allowing players to use different learning rates would be helpful, but we did not pursue that direction since it would break the symmetry in how the players update their strategies.
>
> --- *“In numerical experiments, [...] but less standard.”*
>
> While our analysis does not cover algorithms such as optimistic multiplicative weights, it does go well-beyond Euclidean L2 regularization. We also stress that optimistic projected gradient descent (OMD with Euclidean regularization) is an extremely well-studied algorithm in recent years; we will make sure to stress this point further in our revised version.
>
> --- *“Section 4.1 needs a bit more polishing. [...]"*
>
> We will make sure to further polish Section 4.1.
>
> --- *“Figure 2 is also a bit confusing [...]“*
>
> The x-axis of Figure 2 shows the **utility** of player X, while the y-axis the **utility** of player Y (please see the labels in the plot). We will make sure to also specify that in the caption.
>
> --- *“Some of the numerical details [...]”*
>
> We will transfer many of the numerical details in the appendix.
>
> --- *“One caveat is the perhaps restrictive set of assumptions: two-player games”*
>
> Two-player general-sum games are one of the most fundamental classes of games (please see Appendix A for an overview). Further, while our main result cannot be extended in multiplayer games (Remark 3.7), we still believe that there are interesting classes of games for which it can be extended.
>
> --- *“How is Lemma 3.3 different from standard RVU bound [...]*
>
> The RVU bound is a statement about the regret of each player, while Lemma 3.3 does not involve at all the regrets of either player. Also, notice that the left-hand side of Lemma 3.3 does not involve  a squared norm—unlike the RVU bound. So the RVU bound is quite different from Lemma 3.3. Their proofs also seem quite different to us, apart from minor similarities in some steps of the proof. Perhaps the reviewer could elaborate more on this point.
>
> Regarding intuition about Lemma 3.3, that result is a crucial piece in the proof of our main result, as it connects the path lengths between the two players. Notice that this particular step breaks in multiplayer games; there, it is possible that some of the players are completely “disconnected” from the other players, in which case there is no connection whatsoever between their path lengths.
>
> --- “Last sentence in abstract, [...]”
>
> Efficiency here is meant in terms of regret, which is a standard measure of performance in the literature on learning in games—this is alluded to in the previous sentence of the abstract, but we will make sure to clarify it in the revised version. In particular, when the dynamics do not reach Nash equilibria, the guarantee in terms of the regret is remarkably better than the $O(1)$ guarantee which is possible, for example, in zero-sum games. Naturally, such regret guarantees also translate to welfare guarantees, but unfortunately that only applies in smooth games (see Syrgkanis et al. (2015)). Establishing improved guarantees in terms of the social welfare in general games is an open question.

---

> > ### Author Response · Authors · 2022-08-08
> > **Thank you for the feedback. Have we addressed the concerns?**
> >
> > We thank again the reviewer for the helpful feedback. Given that the discussion period is soon coming at an end, please let us know if we have adequately addressed the concerns raised, and if the reviewer has any further questions.

---

> > ### Comment · Reviewer_6PC8 · 2022-08-08
> > **Thank you for your response**
> >
> > I thank the authors for the response and apologize for the delay in the discussions. I think the authors' remarks about the learning rates mostly address my concerns there---In this paper $\eta=O(\epsilon^2)$ with $\epsilon=\Theta(1)$ still has interesting implications, unlike in standard no-regret bounds where an average regret of $\Theta(1)$ is not that hard to achieve. On the other hand, the authors have thought about large learning rate that does not depend on $\epsilon$ (like other Optimistic Hedge papers e.g. Daskalakis et al. 2021) and concluded it currently seems a hard question.
> >
> > I also liked the new observation pointed out in the response to Reviewer Yw3X, that the theorem can be strengthened by weakening the precondition to "not visiting approximate NE for only a $\delta$-fraction of the iterates" instead of all iterates (with result degrading when $\delta$ gets small). Therefore, while I am still a bit concerned by the restrictiveness of the assumptions overall, I am now more optimistic about the flexibilities / potentials to be extended and thus have increased my score.

---

### Meta-Review · Area_Chair_z6vM · 2022-08-31

**Recommendation:** Accept
**Confidence:** Certain

**Metareview:**

This paper proves a new phenomenon about the Optimistic Mirror Descent (OMD) algorithm in two-player general-sum matrix games (bimatrix games): The iterates either converge to an approximate Nash Equilibrium (NE), or converge to a Strong Coarse Correlated Equilibrium (CCE). This result links and improves over two existing understandings: (1) Convergence to NE is unlikely to be generally achievable by any efficient algorithm due to its PPAD-hardness; (2) Convergence to approximate CCE is achievable by any no-regret algorithm, but it is unclear whether such CCE is a strong CCE (in the present paper’s sense).

Given the phenomenon is not only new but also fundamental and important to the game theory community, I believe it is worth the attention of NeurIPS audience, and thus recommend acceptance.

**Award:**

No

---

### Decision · Program_Chairs · 2022-09-14

Accept